# Self-Adjusting Weighted Expected Improvement for Bayesian Optimization

Carolin Benjamins[1]  Elena Raponi[2,3]  Anja Jankovic[3]  Carola Doerr[3]  Marius Lindauer[1]

[1]Institute of AI, Leibniz University Hannover, Germany
[2]TUM School of Engineering and Design, TU München, Germany
[3]Sorbonne Université, CNRS, LIP6, Paris, France

**Abstract** Bayesian Optimization (BO) is a class of surrogate-based, sample-efficient algorithms for optimizing black-box problems with small evaluation budgets. The BO pipeline itself is highly configurable with many different design choices regarding the initial design, surrogate model, and acquisition function (AF). Unfortunately, our understanding of how to select suitable components for a problem at hand is very limited. In this work, we focus on the definition of the AF, whose main purpose is to balance the trade-off between exploring regions with high uncertainty and those with high promise for good solutions. We propose Self-Adjusting Weighted Expected Improvement (SAWEI), where we let the exploration-exploitation trade-off self-adjust in a data-driven manner, based on a convergence criterion for BO. On the noise-free black-box BBOB functions of the COCO benchmarking platform, our method exhibits a favorable anytime performance compared to handcrafted baselines and serves as a robust default choice for any problem structure. The suitability of our method also transfers to HPOBench. With SAWEI, we are a step closer to on-the-fly, data-driven, and robust BO designs that automatically adjust their sampling behavior to the problem at hand.

## 1 Introduction

Black-box problems are challenging to optimize because we do not have direct access to the underlying structure of the problem landscape. To optimize them, we can sequentially evaluate different points $x$ and use the obtained objective values $f(x)$ to choose which point(s) to evaluate next, but we do not have *a priori* information where to find the most promising regions or how to best trade off exploration of the search space with exploitation of regions that appear to be very promising. Formally, in black-box optimization we want to find the minimum $x^*$ of a given function $f$, $x^* \in \arg\min_{x \in \mathcal{X}} f(x)$, without having access to the function itself other than through the queries. Typical black-box problems occur in engineering or hyperparameter optimization (HPO), where the quality of potential solutions is evaluated via numeric simulations or training machine learning models.

Balancing exploration with exploitation is particularly challenging when we have a low number of available function evaluations in relation to the size of the search space $\mathcal{X}$. A popular approach to address such settings is Bayesian optimization (BO) (Mockus, 1989; Garnett, 2023), often promoted as sample-efficient for expensive black-box optimization. The main idea of BO is to use a probabilistic surrogate model (e.g., a Gaussian Process), iteratively refining an approximation of the problem landscape that guides the optimization process. BO starts with an *initial design* or *design of experiment* (DoE), obtained from sampling strategies, e.g., random sampling, low-discrepancy sequences such as Sobol', or Latin Hypercube design (Brochu et al., 2010). With these initial points, the surrogate model is built to approximate the unknown objective function and capture the uncertainty of the true function value on unobserved points. The *acquisition function* (AF) (a.k.a. *infill criterion*) is a utility function to trade off exploration of underexplored areas and exploitation

of presumably promising ones. The point with the highest acquisition function value is queried. Afterwards, the surrogate model is adjusted with the new observation, and the optimum is updated if the new point improves the target value of the best-so-far observation. These steps are repeated for a given overall optimization budget.

Besides accurate probabilistic surrogate models and the type and size of initial design (Lindauer et al., 2019; Bossek et al., 2020; Cowen-Rivers et al., 2021), the exploration-exploitation trade-off is crucial for successful and efficient optimization. Since the landscape of the black-box optimization problem is unknown, it is a priori unclear which AF should be chosen for the optimization problem at hand. Even worse, since each problem has its unique landscape, we need different exploration-exploitation trade-offs (Benjamins et al., 2022a,b).

Because there are different choices of AFs, e.g., Probability of Improvement (PI) (Kushner, 1964), Expected Improvement (EI) (Mockus et al., 1978), Upper Confidence Bound (UCB) (Forrester et al., 2008), Thompson Sampling (TS) (Thompson, 1933), Entropy Search (Hennig and Schuler, 2012) and Knowledge Gradient (Frazier et al., 2009), selecting a suitable one for the problem at hand with insights on the landscape remains challenging. Furthermore, in the past, the choice of an AF has been considered *static* over the BO process. Prior works suggest that mixed AF-strategies (Hoffman et al., 2011; Kandasamy et al., 2020) or even very simple schedules switching from EI to PI can improve anytime performance of BO; however, for each problem different schedules, incl. static ones, perform best (Benjamins et al., 2022b).

Performance can be improved by selecting an AF-schedule with a meta-learned selector based on the exploratory landscape analysis (ELA) features (Mersmann et al., 2011) of the initial design which factors in the problem at hand (Benjamins et al., 2022a). Nevertheless, this approach has its limitations. First, it requires a large and expensive initial design compared to the overall budget in order to compute the ELA features, and the ideal size of it is unknown (Belkhir et al., 2016). Second, the selector is trained for a specific budget, and it is unclear how it transfers to other dimensions, optimization budgets, or initial designs.

In this work, we instead aim for a *self-adjusting yet simple* approach to adapt the exploration-exploitation trade-off in a data-driven way throughout the optimization process. For this, we propose to adaptively set the weight $\alpha$ of Weighted Expected Improvement (WEI) (Sobester et al., 2005) in an online parameter control fashion (Karafotias et al., 2015; Doerr and Doerr, 2020). Depending on how we parametrize WEI, we can be more explorative, recover EI, or lean towards a modulated, exploitative PI. The crucial questions to answer here are (i) *When* should we adjust $\alpha$? and (ii) *How* should we adjust $\alpha$?

We propose a new method, dubbed Self-Adjusting Weighted Expected Improvement (SAWEI). Inspired by a termination criterion for BO (Makarova et al., 2022), we adjust the weight $\alpha$ whenever BO tends to converge, indicated by the Upper Bound Regret (UBR). We adjust $\alpha$ opposite to the dominant search attitude, either towards exploration or exploitation. The key mechanism behind SAWEI is illustrated in Figure 1. We demonstrate the effectiveness of our method SAWEI on the BBOB functions of the COCO benchmark (Hansen et al., 2020) and on tabular benchmarks from HPOBench (Eggensperger et al., 2021) against baselines of established AFs and previously proposed handcrafted AF-schedules for $\alpha$.

## 2 Related Work

One line of works directly focuses on improving AFs (Qin et al., 2017; Balandat et al., 2020; Volpp et al., 2020). To overcome the fact that EI can sometimes be too exploitative, Qin et al. (2017) uniformly sample one of the two most promising points instead of always choosing the most promising one according to EI. Balandat et al. (2020) offer efficient implementation of Monte-Carlo AFs (no closed-form solution available) as well as a one-shot formulation of the Knowledge Gradient. A different approach is to meta-learn a neural AF via Reinforcement Learning to achieve better sample-efficiency on downstream tasks (Volpp et al., 2020). A different line of work is

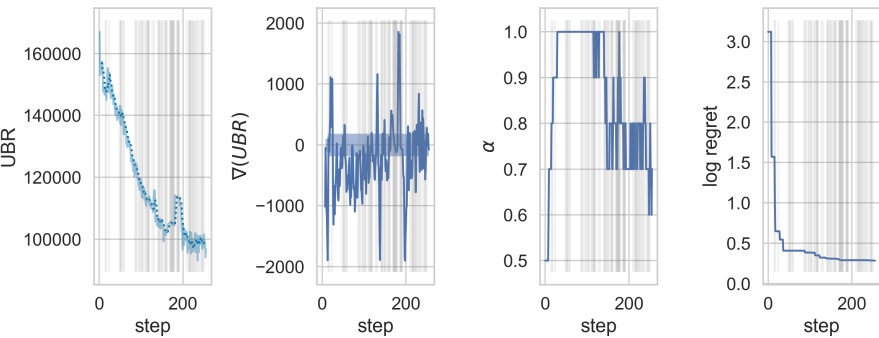

Figure 1: With SAWEI we self-adjust the exploration-exploitation trade-off parameter $\alpha$ based on the Upper Bound Regret (UBR) (left). Whenever the gradient of UBR (2nd left) becomes approximately 0 (marked by vertical lines), we adjust $\alpha$ (2nd right), further reducing the log regret (right). BBOB F20, 8d.

concerned with combining different AFs, e.g., by building a portfolio of AFs (EI, PI, UCB with different hyperparameter settings) and then using an online multi-armed bandit strategy to assign probabilities of which AF to use at which step, called GP-Hedge or Portfolio Allocation (Hoffman et al., 2011). Their work indicates that the performance of Portfolio Allocation highly varies with the number of arms and their respective hyperparameter settings. Similarly to Portfolio Allocation, Kandasamy et al. (2020) update weights of their portfolio (UCB, EI, TS (Thompson, 1933), Top-Two Expected Improvement (TTEI) (Qin et al., 2017)) in an online manner. They do not include PI as they observe it exhibits inferior performance compared to other single static AFs. In addition, robust versions of EI, PI, and UCB can be combined to a multi-objective AF combining the strengths of the individual ones (Cowen-Rivers et al., 2021). In this work, we take a step back and ask ourselves what we could achieve by employing a simplistic approach of self-adjusting the exploration-exploitation trade-off of WEI.

It has also been shown in other optimization-related areas that dynamic choices are beneficial in terms of performance, e.g., in evolutionary computation (Karafotias et al., 2015; Doerr and Doerr, 2020), planning (Speck et al., 2021) and deep learning (Adriaensen et al., 2022). Recently, the introduction of Dynamic Algorithm Configuration (DAC) (Biedenkapp et al., 2020) underlines the potential of employing dynamic schedules (as opposed to selecting algorithm components *on the fly*, as is usually done in evolutionary computation (Hansen et al., 2003)).

Related to that, also setting the weight $\alpha$ of WEI has been investigated. Sobester et al. (2005) propose to cycle through $\alpha \in \{0.1, 0.3, 0.5, 0.7, 0.9\}$ to pulse from exploring to exploiting. This idea is based on the suggestion to cycle through global-local balances during the search (Gutmann, 2001). However, this heuristic is oblivious to the current state of the search. Another line of work proposes to simply query WEI $n$ times with $n$ different values of $\alpha$ in parallel (Liu et al., 2018) with the drawback of potentially uninformative function evaluations. The weights for the exploration and exploration terms in WEI can also be set via rewards obtained by calculating the accuracy of the surrogate model (Xiao et al., 2012, 2013). However, the definition of the rewards is lacking and their method needs to be reset from time to time for the case when the exploration term causes to repeatedly propose the same configuration.

## 3 Self-Adjusting Weighted EI

In our method, the Self-Adjusting Weighted Expected Improvement (SAWEI), we adaptively set the weight $\alpha \in [0, 1]$ of the Weighted Expected Improvement (WEI) to steer the exploration-exploitation

trade-off. WEI (Sobester et al., 2005) is defined as:

$$WEI(\mathbf{x}; {\color{red}\alpha}) = {\color{red}\alpha} \underbrace{z(\mathbf{x})\hat{s}(\mathbf{x})\Phi\left[z(\mathbf{x})\right]}_{\text{exploitation-driven}} + {\color{red}(1-\alpha)} \underbrace{\hat{s}(\mathbf{x})\phi\left[z(\mathbf{x})\right]}_{\text{exploration-driven}} \tag{1}$$

with $z(\mathbf{x}) = (f_{\min} - \hat{y}(\mathbf{x}))/\hat{s}(\mathbf{x})$, $f_{\min}$ being the lowest observed function value, $\hat{y}(\mathbf{x})$ and $\hat{s}(\mathbf{x})$ the predicted mean and standard deviation from the surrogate model, and $\phi$ and $\Phi$ being the PDF and CDF of a Gaussian distribution, respectively. The $\alpha$ coefficient weighs the exploration and exploitation terms. For example, $\alpha = 0.5$ recovers standard EI (Mockus et al., 1978) and $\alpha = 1$ has a similar behavior as $PI(\mathbf{x}) = \Phi\left[z(\mathbf{x})\right]$ (Kushner, 1964). With $\alpha = 0$ we only utilize the exploration term, but this does not equal pure exploration or complete randomness.

**When To Adjust**. In order to be able to set $\alpha$ adaptively, we need an indicator of the progress of the optimization. Recently, Makarova et al. (2022) proposed a termination criterion to stop BO for hyperparameter optimization. If the Upper Bound Regret (UBR) falls under a certain threshold, they terminate. UBR estimates the true regret at iteration $k$ by:

$$\text{UBR}(G_k; \mathcal{X}) = r_k := \min_{\mathbf{x} \in G_k} \text{UCB}_k(\mathbf{x}) - \min_{\mathbf{x} \in \mathcal{X}} \text{LCB}_k(\mathbf{x}) \tag{2}$$

with $G_k$ being the history of all evaluated points, $\mathcal{X}$ being the entire search space, and LCB and UCB being the lower and upper confidence bound, e.g., $UCB(x) = \mu_t(x) + \beta_t \sigma_t(x)$ and $LCB(x) = \mu_t(x) - \beta_t \sigma_t(x)$, respectively. The first term of UBR estimates the worst-case function value of the best-observed point, a.k.a. the incumbent, and the second term is the lowest function value across the whole search space. This means the smaller the gap between both terms becomes, the closer we are to the asymptotic function value *under the current settings of the optimizer*. We empirically show that the UBR indeed changes after we change the acquisition function during the optimization in Appendix C, supporting our intuition. The UBR does not directly operate on function values, but UCB and LCB are computed on the surrogate model instead. Instead of using UBR to stop the optimization process, it serves as an indicator for us when to adjust components, i.e., update the value of $\alpha$.
**Our rule is**: When the gradient of UBR over the last $n$ steps becomes close to 0, we adjust the exploration-exploitation attitude with $\alpha$. The sensitivity to the gradient is controlled by our hyperparameter $\epsilon$.

**How to Adjust**. The remaining question is *how* to adjust $\alpha$, by how much and into which direction. We propose a rather simple, yet effective additive change by $\Delta_\alpha$. Our intuition is to set $\alpha$ opposite to the current *search attitude*, since the current search attitude led to convergence of the optimization. The term search attitude describes the current search behavior, whether the acquisition function is more explorative or more exploitative. We set $\Delta_\alpha = 0.1$ to allow for gradual changes. We determine the sign of $\Delta_\alpha$ by the recent search attitude: depending on whether the exploration-term $a_{\text{explore}}$ or exploitation-term $a_{\text{explore}}$ of Equation (1) is larger for the last selected point $\mathbf{x}_{\text{next}}$, the current search attitude was either steered more for exploring or exploiting, respectively. The terms are the summands of WEI and defined as follows:

$$a_{\text{explore}}(\mathbf{x}_{\text{next}}) = \hat{s}(\mathbf{x}_{\text{next}})\phi\left[z(\mathbf{x}_{\text{next}})\right] \tag{3}$$

$$a_{\text{exploit}}(\mathbf{x}_{\text{next}}) = z(\mathbf{x}_{\text{next}})\hat{s}(\mathbf{x}_{\text{next}})\Phi\left[z(\mathbf{x}_{\text{next}})\right] \tag{4}$$

We use $a_{\text{exploit}} = \Phi\left[z(\mathbf{x}_{\text{next}})\right]$, omitting $z(\mathbf{x}_{\text{next}})\hat{s}(\mathbf{x}_{\text{next}})$, which is equal to PI.[1] Please note that we only do this for determining the search attitude. Now if the exploration term is bigger than the

---

[1]Empirically, both methods perform almost equivalent for BBOB but not for HPOBench, see Appendix B. We conjecture that original $a_{\text{exploit}}$ is less exploitative than the original PI. Since we look for a strong (global) signal on how exploitative a point was, we opted for PI instead of the WEI term.

exploitation term, i.e., $a_{\text{explore}} > a_{\text{exploit}}$, the current search attitude is exploration. We inspect the attitude and adjust $\alpha$ in the *opposite* direction, to provide a chance for more exploration or exploitation in contrast to the currently dominating attitude.

**SAWEI in a Nutshell.** We illustrate and summarize our method SAWEI in Figure 1 and in Algorithm 1. Our goal is to adjust the exploration-exploitation trade-off based on the current search attitude whenever the Upper Bound Regret (UBR) converges. SAWEI enhances the standard BO pipeline by calculating the UBR in each iteration and by tracking the search attitude via the exploration term and the exploitation term of WEI. First, we define and evaluate the initial design and train our surrogate model (Line 1). Then, as long as we have function evaluations left (Line 2), we query the acquisition function (here Weighted Expected Improvement (WEI)) for the next point to be evaluated (Line 3). Meanwhile, we track the search attitude with the exploration and exploitation terms of WEI (Line 4, see Equation (3)). The function is evaluated as usual with the proposed point and we update our history and our surrogate model (Lines 5-7). Now we calculate the UBR estimating the gap to the true regret based on the history of evaluated points and the search space (Line 8). We smooth the history of UBR with moving interquartile mean (IQM) (25 %-75 % quartiles) with a window size of 7 (Lines 9-10, `smooth_with_iqm`). Based on this smoothed version, we check whether UBR has converged, i.e., the gradient of UBR is close to 0 (Line 11). In more detail, we signal time to adjust when the last absolute gradient is close to 0 with an absolute tolerance of $\epsilon$ times the last observed maximum of the absolute gradient. If it is the case, we adjust the weight $\alpha$ of WEI based on the search attitude (Line 12). The search attitude is calculated with the exploration and exploitation terms of WEI.

---

**Algorithm 1** Bayesian Optimization with Self-Adjusting Weighted Expected Improvement (SAWEI)

---

**Require**: Initial weight of WEI $\alpha = 0.5$, history of evaluated points $G = \emptyset$, history of regret estimates/UBR $R$, surrogate model $\mathcal{M}$, function to optimize $f$

  1: Evaluate initial design and train surrogate model $\mathcal{M}$
  2: **while** Optimization Budget Not Exhausted **do**
  3:     $x_{\text{next}} \leftarrow \text{WEI}(\mathcal{M})$                                        ▷ Propose next configuration to evaluate
  4:     $a_{\text{explore}}, a_{\text{exploit}} \leftarrow \text{WEI}(x_{\text{next}})$        ▷ Get summands of $\text{WEI}(x_{\text{next}})$ before $\mathcal{M}$ is trained
  5:     $y \leftarrow f(x_{\text{next}})$                                                    ▷ Evaluate function
  6:     $G \leftarrow G \cup \{x_{\text{next}}\}$                                              ▷ Update history
  7:     Train surrogate model $\mathcal{M}$
  8:     $r \leftarrow \text{UBR}(G, \mathcal{X})$                            ▷ Upper Bound Regret (UBR) estimate, Equation (2)
  9:     $R \leftarrow R.\text{append}(r)$
 10:     $\bar{R} \leftarrow \text{smooth\_with\_iqm}(R)$                        ▷ Smooth rugged signal with moving IQM
 11:     **if** $\nabla \bar{R} \approx 0$ **then**                                          ▷ Check if UBR converged
 12:         $\alpha \leftarrow \text{adjust}(\alpha, a_{\text{explore}}, a_{\text{exploit}})$   ▷ Adjust exploration-exploitation based on attitude
 13:     **end if**
 14: **end while**

---

## 4 Experiments

In our experiments, we empirically evaluate our method SAWEI on different benchmarks and compare it to baselines from the literature and handcrafted ones. We benchmark the algorithms on the BBOB functions from the COCO problem suite (Hansen et al., 2020) and on HPOBench (Eggensperger et al., 2021). Our implementations are built upon the BO tool SMAC3 (v2.0.0b1) (Lindauer et al., 2022). We use a standard GP as configured in SMAC's BlackBoxFacade and SMAC optimizes the acquisition function with a combination of local and random search which also applies to minimizing LCB in Equation (2) for calculating the UBR. We set $\beta_t = 2\log(dt^2/\beta)$, $\beta = 1$

| | |
|---|---|
| WEI($\alpha = 0$) (Explore) | $\alpha = 0.0$ |
| WEI($\alpha = 0.5$) (EI) | $\alpha = 0.5$ |
| WEI($\alpha = 1$) (modulated PI) | $\alpha = 1.0$ |
| WEI($\alpha = 0.5$) $\rightarrow$ WEI($\alpha = 1$) (Steps) | 5 steps |
| WEI($\alpha = 1$) $\rightarrow$ WEI($\alpha = 0.5$) (Steps) | 5 steps |
| EI $\rightarrow$ PI | switch after 25 % |
| EI $\rightarrow$ PI | switch after 50 % |
| EI $\rightarrow$ PI | switch after 75 % |
| Gutmann-Sobester Pulse (Gutmann, 2001; Sobester et al., 2005) | Cycle $\alpha \in [0.1, 0.3, 0.5, 0.7, 0.9]$ |
| Portfolio Allocation (Hoffman et al., 2011) | - |

Table 1: Baselines.

for UCB/LCB as done in SMAC following the original UCB (Srinivas et al., 2010). The code is available at `https://github.com/automl/SAWEI`. The exact setting for our method is $\epsilon = 0.1$ and adding or subtracting $\Delta\alpha = 0.1$. We set our convergence check horizon to $n = 1$, i.e., we check whether the last gradient is close to 0. We validate our hand-crafted settings an ablation study in Section 4.1.

Our evaluation protocol repeats the optimization 10 times with different random seeds and calculates the interquartile mean (IQM) across seeds to robustly estimate the regret per function. For each schedule, we then determine the rank for each of the 24 BBOB functions and compute the global rank across functions. For the rank table, we aggregate the ranks across the single tasks per schedule with the IQM. In the plots over optimization steps, we show the mean and 95 % confidence interval across all the functions.

**BBOB**. For the 24 noiseless, synthetic BBOB functions (Hansen et al., 2020) we set the dimensionality to 8, the budget of the initial design to 24 function evaluations (FEs), and the budget for the surrogate-based optimization to 256 FEs. We optimize the first three instances of each function. In BBOB, the instances are obtained by scaling, shifting, and rotating the base function (hence preserving the problem structure but changing the embedding).

**HPOBench**. We evaluate all methods on the tabular machine learning benchmarks from HPOBench (Eggensperger et al., 2021). To this end, we randomly selected eight tasks from the OpenML dataset (Casalicchio et al., 2017; Feurer et al., 2021) and optimize a Random Forest, MLP, SVM, Logistic Regression, and XGBoost. We allow an initial design of 15 FEs and a BO-based optimization budget of 100 FEs. For each FE, we average the metric over the five available seeds.

**Baselines**. We compare our data-driven, self-adjusting method SAWEI to (i) the well-established best practice of simply using a single AF (EI, PI, and LCB) and (ii) hand-designed schedules of $\alpha$, see Table 1. We start with static schedules of $\alpha \in \{0, 0.5, 1\}$, either more exploring, EI, or more exploiting. Further, we define a schedule from EI (WEI($\alpha = 0.5$)) to modulated PI (WEI($\alpha = 1$)), and vice versa, as a step function with 5 steps. In addition, we compare to hard switches from EI to PI (Benjamins et al., 2022a) as well as the Gutmann-Sobester pulse cycling through $\alpha$ (Gutmann, 2001; Sobester et al., 2005). We also include Portfolio Allocation (Hoffman et al., 2011) and use their portfolio of nine acquisition functions consisting of different parametrizations of UCB, PI, and EI.

## 4.1 Results

**BBOB**. Our method SAWEI ranks among the first based on final performance (cf. Figure 5a), which is very similar to dynamic baselines going from EI ($\alpha = 0.5$) to the modulated PI ($\alpha = 1$). One drawback of the hand-designed schedules is that the optimization budget needs to be defined

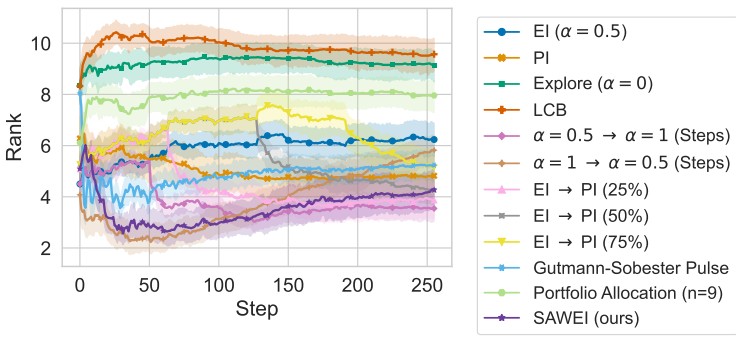

Figure 2: Ranks per Step on BBOB

beforehand, whereas our method is self-adjusting and is oblivious of the total budget. Surprisingly, the modulated PI is comparatively strong and performs better than EI, suggesting that the BBOB landscapes require a higher percentage of exploitation. SAWEI also exhibits a favorable anytime performance, making it a consistent and robust default choice, see Figure 2. Schedules dominating SAWEI only do so for a portion of the optimization, hence they are not consistent. Confirming results from Benjamins et al. (2022b), the effect of switching from EI to PI can be clearly seen as a boost in the ranks. On BBOB, the generally well-performing schedules involve PI which our method can easily mimic. SAWEI finds a suitable transition from exploring to exploiting per-run.

In general, the tendency of the $\alpha$-schedules traversed by SAWEI is moving from exploration to exploitation. Often, we can observe a decrease, a change to more exploration again, after some iterations. On one BBOB function, the multi-modal Schwefel function (F20) with weak global structure (Figure 3a), SAWEI manages to efficiently transform from EI ($\alpha = 0.5$, higher explorative attitude) to modulated PI ($\alpha = 1$) with an exploitative attitude. At the end of the optimization, when the basin was already discovered, SAWEI decreases $\alpha$ to more exploration to explore the surroundings. We can also clearly observe the effect of the hand-designed switching (EI $\rightarrow$ PI (x %)) in the sharp bends downwards in the log regret and upwards in the UBR, although SAWEI discovers a more suitable point and can change its attitude again. On Katsuura, which is highly multi-modal and has weak global structure (Figure 3c), SAWEI increases $\alpha$ more slowly to exploitation, presumably because of the highly rugged landscape, see Figure 3d. Also here, SAWEI discovers the boost from changing from exploration to exploitation. If we look closely we can see that the Upper Bound Regret jumps up after the switch happened for the switching schedules (EI to PI) which is an indication of the adequacy of UBR as a state descriptor. All schedule plots for each BBOB function, as well as the box plots of the final log regrets can be found in Appendix D.

**HPOBench**. On HPOBench we see that SAWEI also has a favorable anytime performance, see Figure 4, and ranks among the first for the final log regret (Figure 5a). It is on par with Explore ($\alpha = 0$), and they are directly followed by Portfolio Allocation and EI. The supremacy of the exploratory schedules is quite surprising, given the simplicity commonly attributed to response landscapes in HPO (Pushak and Hoos, 2018). We will investigate this further in our future work. With a closer look at the schedules, we see the general trend to start from EI ($\alpha = 0.5$) and go to Explore ($\alpha = 0$) which is the complete opposite of the BBOB behavior. Boxplots of the final log regret and all plots with log regret, UBR and $\alpha$ over time can be found in Appendix E.

**Comparison of BBOB and HPOBench**. In summary, we observe that the optimal schedule and search behavior vary on two levels. First, for a given problem type, the optimal schedule varies across the single tasks. Second, the search behavior depends on the type of problem, whether we optimize synthetic functions in BBOB or find optimal hyperparameters for machine learning models in HPOBench. SAWEI mimics the strategy fitting best to the problem at hand and exhibits

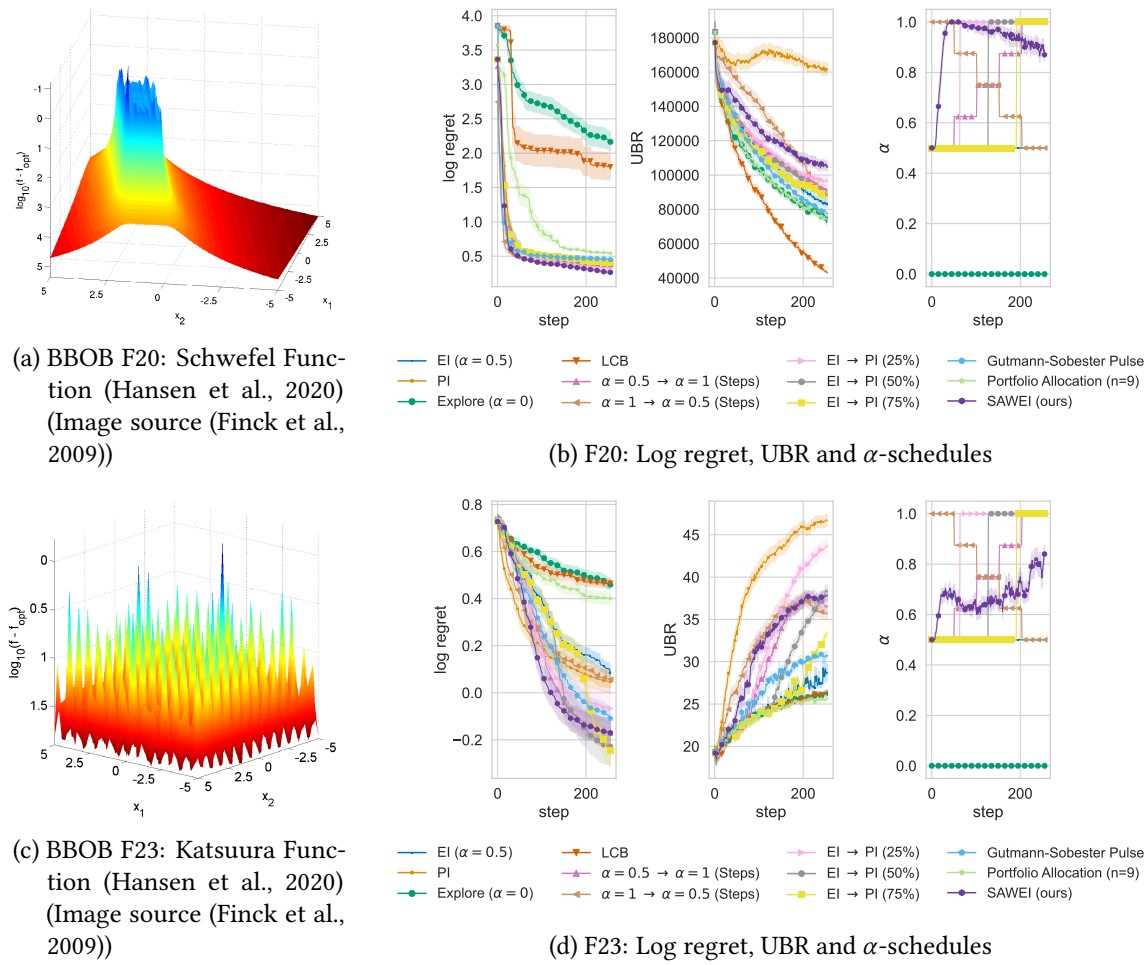

(a) BBOB F20: Schwefel Function (Hansen et al., 2020) (Image source (Finck et al., 2009))

(b) F20: Log regret, UBR and $\alpha$-schedules

(c) BBOB F23: Katsuura Function (Hansen et al., 2020) (Image source (Finck et al., 2009))

(d) F23: Log regret, UBR and $\alpha$-schedules

Figure 3: BBOB Functions F20 and F23 with different performance and behavior indicators

the most favorable rank distribution *across domains*, see Figure 5a. BBOB in general requires more exploitation and HPOBench more exploration which is visible prominently in two ways. First, PI performs better on BBOB than on HPOBench and EI vice versa. Second, SAWEI's trajectories of $\alpha$ are contrary on BBOB and HPOBench (see Figure 5b) and thus adjust to the required search attitude.

**Ablation on BBOB.** We perform an ablation study to assess the sensitivitiy of our method to its hyperparameters. In particular, we vary $\Delta\alpha \in \{0.05, 0.1, 0.25\}$, i.e., the amount to add or subtract to our current weight $\alpha$. In addition, we can track the attitude in different ways: either just considering the last step (`last`), or accumulating the terms until the last point where the best configuration (the incumbent) changed (`inc. change`) or until the last adjustment happened (`last adjust`). In the latter cases, $a_{\text{explore}}$ and $a_{\text{exploit}}$ become sums. This hyperparameter defines the convergence check horizon $n$, which is varied during the run for the latter two options. Finally, we vary the sensitivity to the gradient of UBR by the width of the tolerance band when compared to 0: $\epsilon \in \{0.05, 0.1, 0.5, 1\}$. The bigger $\epsilon$, the more often we switch. We evaluate all 36 combinations on all 24 BBOB functions with 10 seeds and 1 instance on 8 dimensions and assess the hyperparameter importance with fANOVA (Hutter et al., 2014). We normalize the log regret for each BBOB function and use this as the performance metric.

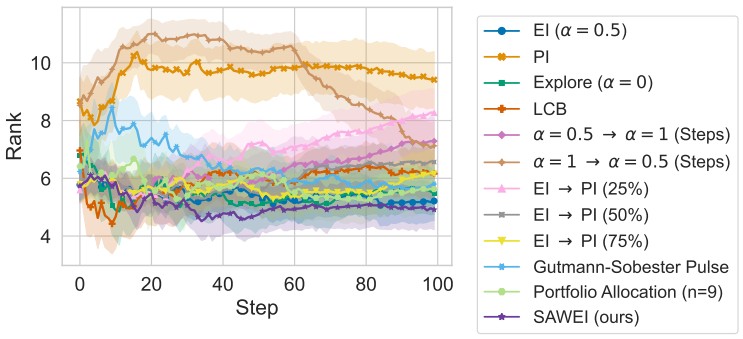

Figure 4: Ranks per Step on HPOBench

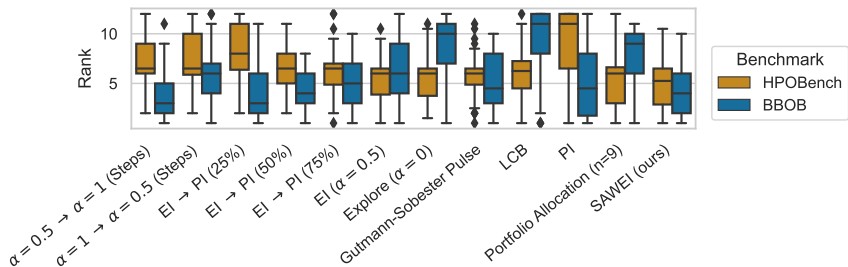

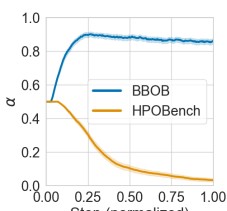

(a) Our method SAWEI has the most favorable rank distribution *across domains*. That is, even though on different benchmarks other schedules perform on par, their suitability highly differs depending on the benchmark. Ranks are computed on final performance.

(b) $\alpha$ traversed by SAWEI which adapts to the different benchmarks.

Figure 5: Comparison of BBOB and HPOBench

We show the marginals of each hyperparameter in Figure 6. The sensitivity $\epsilon$ to the gradient has a slight tendency to 0.05 but the overall differences are small, we argue that the exact timing of the signal to adjust is less important. In addition, setting the granularity of $\Delta\alpha$ is quite robust to the exact setting. In contrast, tracking the attitude has a tendency to favor checking the exploration/exploration terms until the last adjust. It is likely that on other benchmarks the importances might change and our default of $\epsilon = 0.1$, track_attitude $=$ last, $\Delta\alpha = 0.1$ proves to be a robust one.

## 5 Limitations and Future Work

Our method SAWEI introduces a slight overhead due to the need to optimize Lower Confidence Bound (LCB) for computing UBR in each iteration. Everything that follows, namely deciding whether and how to adjust $\alpha$, is negligible in terms of computational cost.

In our analysis, we did not experiment with the initial value of $\alpha$, which may not be optimal for every tested function. Also, our method does not allow jumps or resetting $\alpha$, which could also be beneficial. In this context, defining $\alpha$ directly as a function of the exploration/exploitation terms of WEI could be a way to allow more flexibility.

One limitation is that so far we have only combined EI and PI. Our approach can easily be extended to *any* linear combination of two acquisition functions. Moreover, we can combine SAWEI with Dynamic Algorithm Configuration (DAC) (Biedenkapp et al., 2020) to learn policies of $\alpha$ across instances and tasks. More generally, we strongly believe that meta-learning and self-adjustment should go hand in hand, another topic to be explored in future work. Building on the work

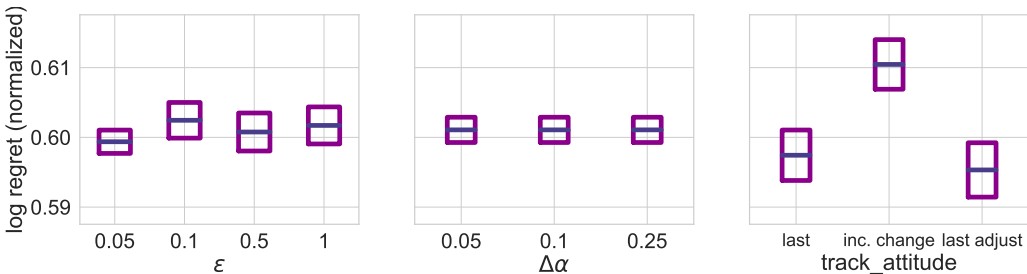

Figure 6: Marginal performances (mean and standard deviation) of SAWEI's hyperparameters

by Benjamins et al. (2022a), one could consider to warmstart SAWEI using meta-models utilizing ELA features (Mersmann et al., 2011). Future work, and a current limitation, is the investigation of more domains as the domains show large variations. Finally, we believe that also other components of BO like the surrogate model could benefit from self-adjusting choices.

## 6 Conclusions

Through a self-adjusting choice of the acquisition function in Bayesian Optimization, we aim to benefit from two main levers: (1) an automated identification of the AF best suitable for the unknown task at hand (e.g., while PI performs better than EI on BBOB, it is the other way around for HPO problems), and (2) an adjustment to the different needs during the optimization process.

Our method SAWEI uses the convergence of Upper Bound Regret (UBR) as a criterion for *when* to adjust its parametrized acquisition function. SAWEI proves to achieve promising performance on two classic benchmark suites, BBOB and HPOBench, outperforming the static EI and PI AFs. It is hence able to achieve both goals, (1) and (2), listed above. It furthermore does not only achieve good final ranks, but also exhibits a favorable anytime performance on both suites.

As a side result of our study, we observe that the general trends in BBOB and HPOBench are orthogonal to each other: while SAWEI generally traverses from EI (exploration) to a modulated PI (exploitation) for BBOB, it moves from EI to even more exploration on HPOBench. This demonstrates the need for flexible, on-the-fly-adjustment of BO components.

**Broader Impact Statement**: After careful reflection, the authors have determined that this work presents no notable negative impacts on society or the environment, since it presents a foundational approach without any concrete application at hand.

**Acknowledgements**. The authors gratefully acknowledge the computing time provided to them on the high-performance computers Noctua2 at the NHR Center PC2 under the project hpc-prf-intexml. These are funded by the Federal Ministry of Education and Research and the state governments participating on the basis of the resolutions of the GWK for the national high performance computing at universities (www.nhr-verein.de/unsere-partner). Carolin Benjamins and Marius Lindauer acknowledge funding by the German Research Foundation (DFG) under LI 2801/4-1. Elena Raponi acknowledges funding by the PRIME programme of the German Academic Exchange Service (DAAD) with funds from the German Federal Ministry of Education and Research (BMBF).

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

Figure 7: Search Attitude Variation

## A  Hardware and Runtime

**Acknowledgements.**  All experiments are conducted on a CPU cluster with 990 nodes with AMD Milan 7763 CPUs. The compute time for the BBOB 8d functions was 45 min each so 14 040 h = 585 d in total on CPU (including ablation). The compute time for the HPOBench was 90 sec each so 288 h = 12 d in total on CPU.

## B  Search Attitude

We determine the search attitude based on the exploration-term of WEI and PI ($\Phi\left[z(\mathbf{x}_{\text{next}})\right]$, Section 3). Originally we compared the exploration-term with the exploitation-term of WEI, the latter being a modified version of PI ($z(\mathbf{x}_{\text{next}})\hat{s}(\mathbf{x}_{\text{next}})\Phi\left[z(\mathbf{x}_{\text{next}})\right]$). We evaluate both versions on BBOB (all 24 functions, 8d, 3 instances, 10 seeds, like in main) and HPOBench (5 models on 8 tasks, 10 seeds, like in main). In Figure Figure 7a on BBOB, we see that the current version (SAWEI (ours)) achieves slightly lower log regret than the one using the modified PI term (SAWEI (modPI)) but otherwise the distributions seem very similar. On HPOBench, the log regret of SAWEI (modPI) is drastically worse than for SAWEI (ours). Please note that we denote the optimum log regret of $\log(0)$ by $-10\,000$. This can be explained by the traversed $\alpha$, see Figure 7b. SAWEI (modPI) adjusts $\alpha$ to exploitation where exploration is required. In addition, SAWEI (modPI) is not able to reduce $\alpha$ again for BBOB.

## C UBR Intuition

The Upper Bound Regret (UBR) can be used to stop BO (Makarova et al., 2022). This means the UBR signalizes whether it is worth to continue optimization. We add our intuition that this holds for the *current optimizer settings*. This is empirically supported by observing the UBR for the switching policies (EI to PI) where we see sharp bends in the UBR after switching, see Figure 8. In our case "current setting" implicitly describes the search attitude whether it is exploring or exploiting. Therefore we can use the UBR to signal when we should change our search attitude.

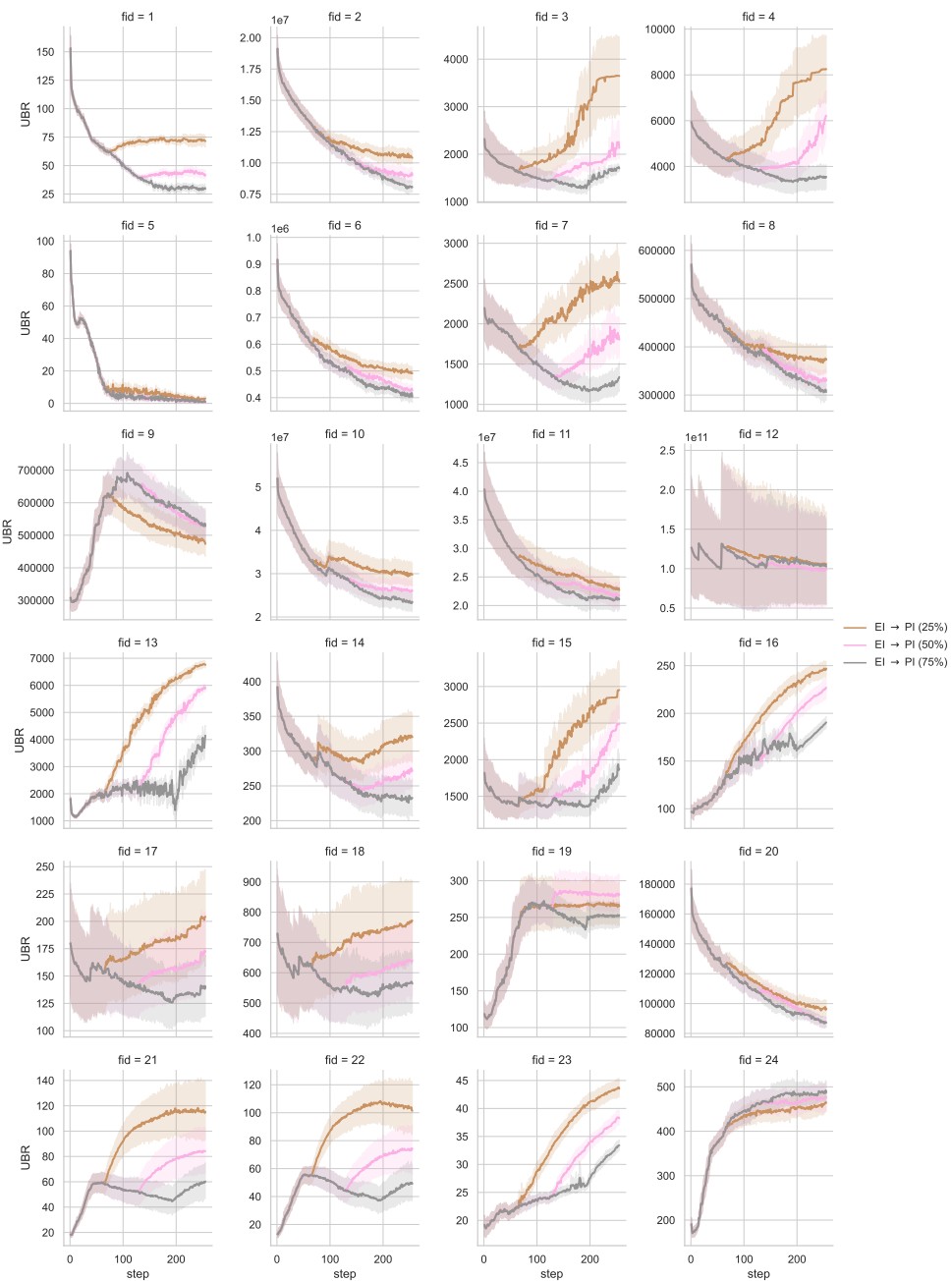

Figure 8: Effect in the Upper Bound Regret (UBR) after changing optimizer settings. Here we switch the acquisition function from EI to PI. BBOB functions, 8d, 10 seeds, 3 instances.

# D  BBOB Results

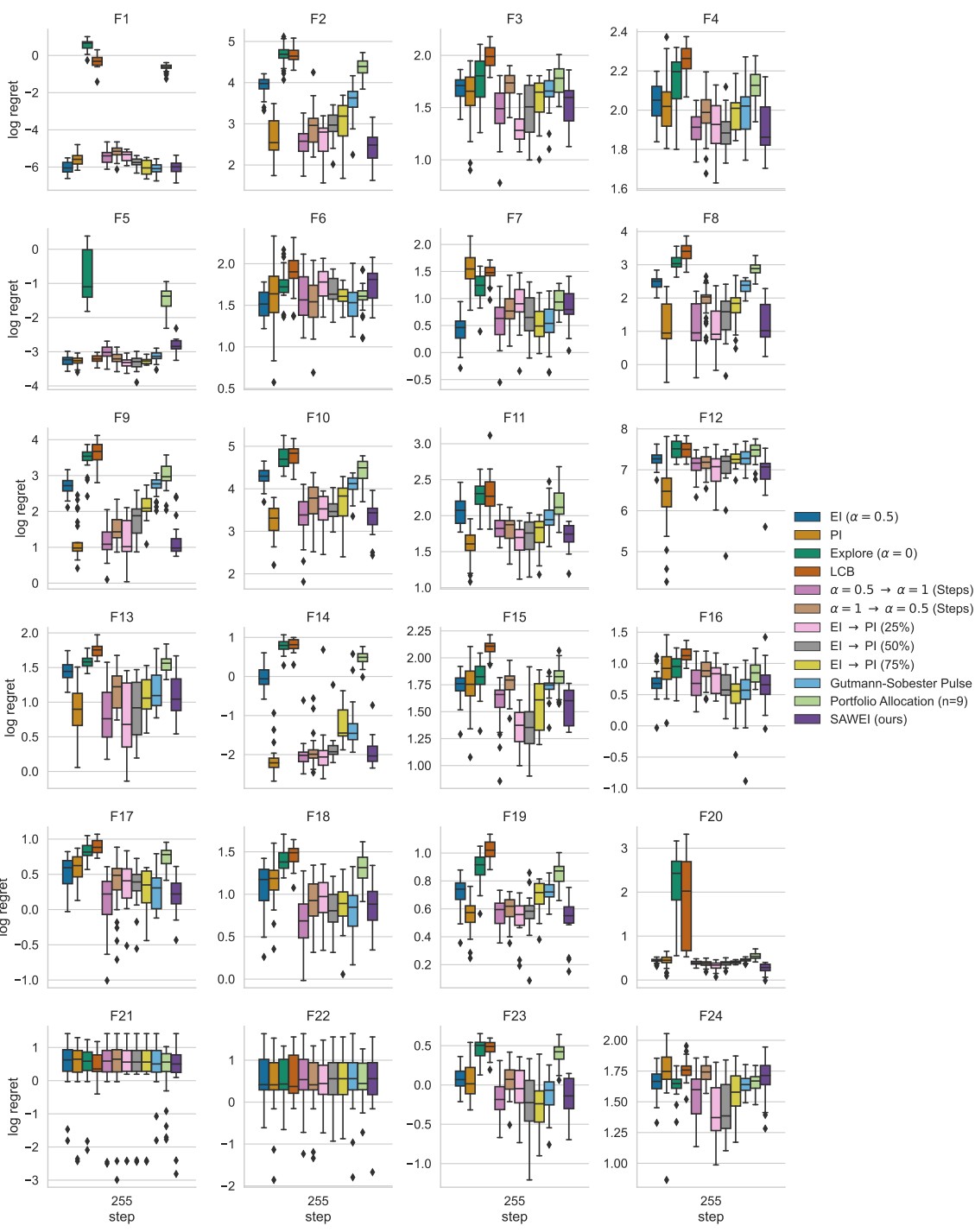

Figure 9: Final log regret on BBOB (8d, 10 seeds, 3 instances)

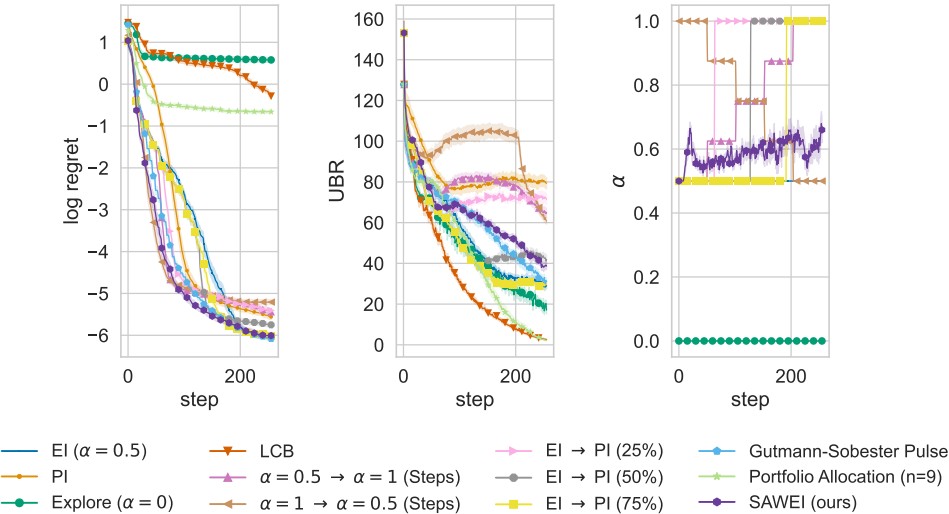

Figure 10: BBOB Function 1

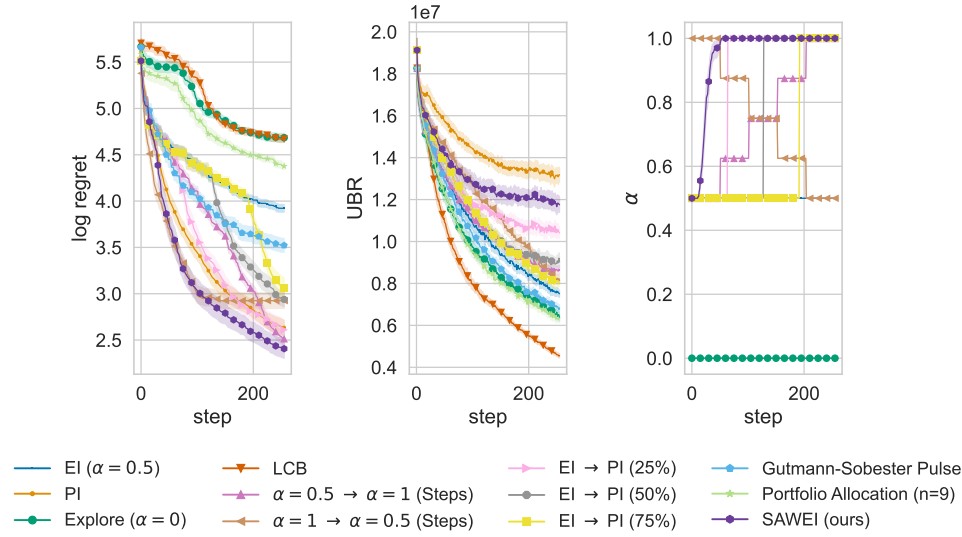

Figure 11: BBOB Function 2

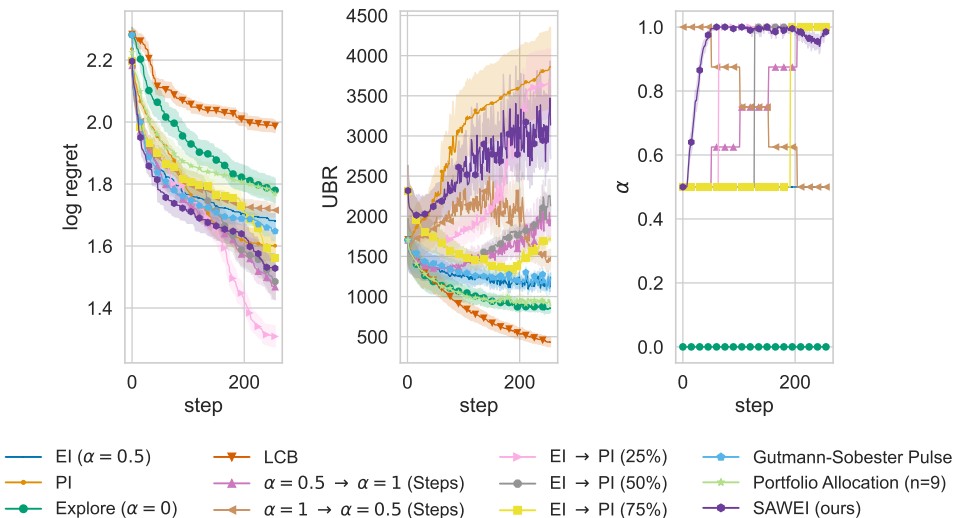

Figure 12: BBOB Function 3

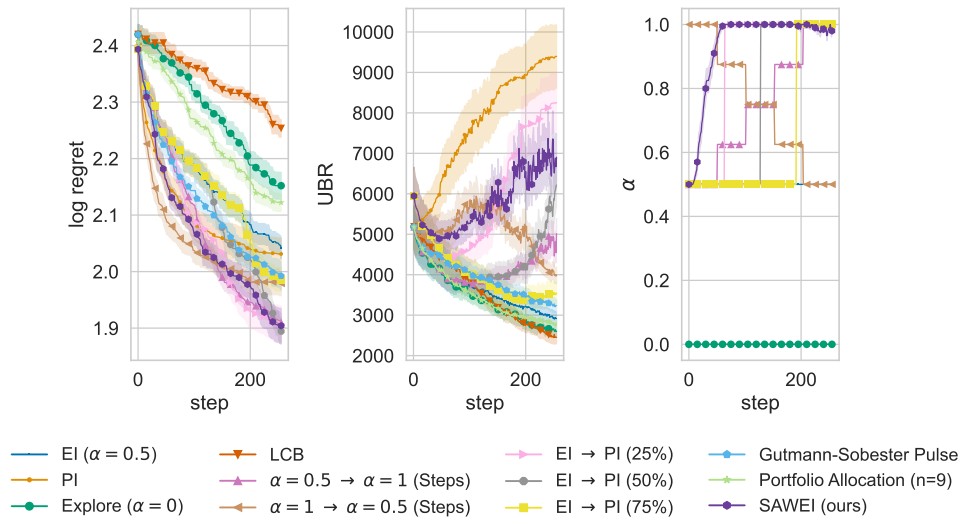

Figure 13: BBOB Function 4

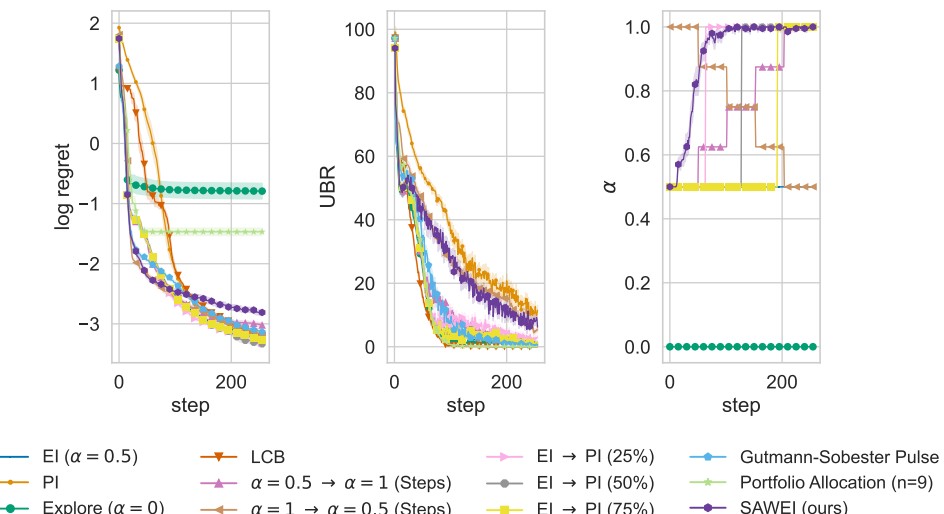

Figure 14: BBOB Function 5

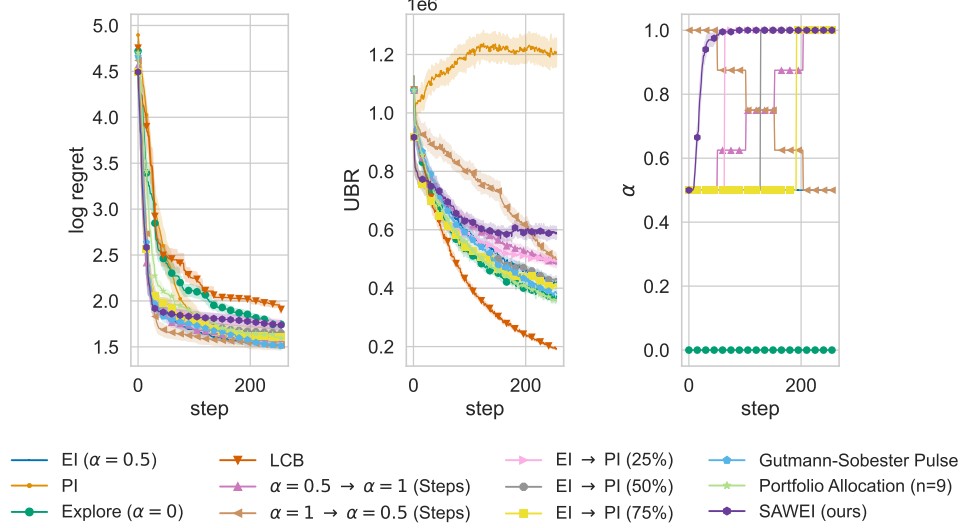

Figure 15: BBOB Function 6

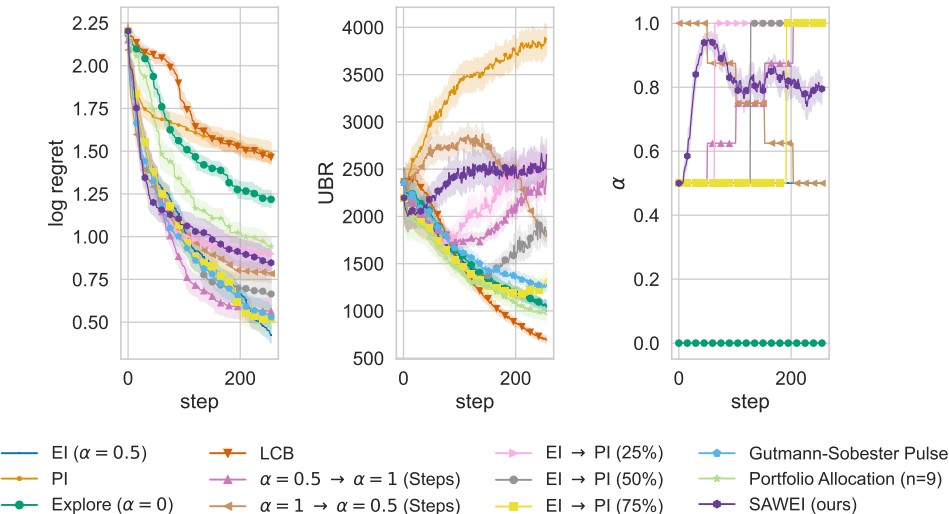

Figure 16: BBOB Function 7

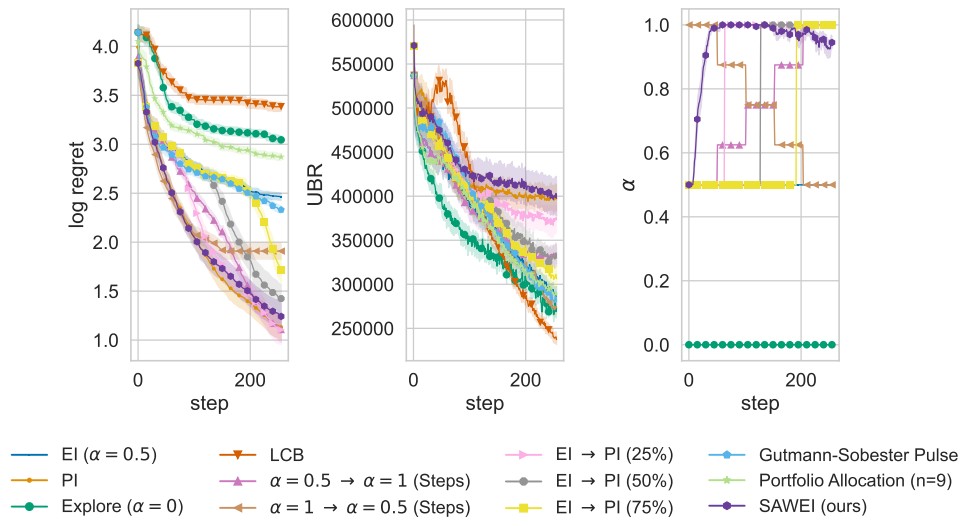

Figure 17: BBOB Function 8

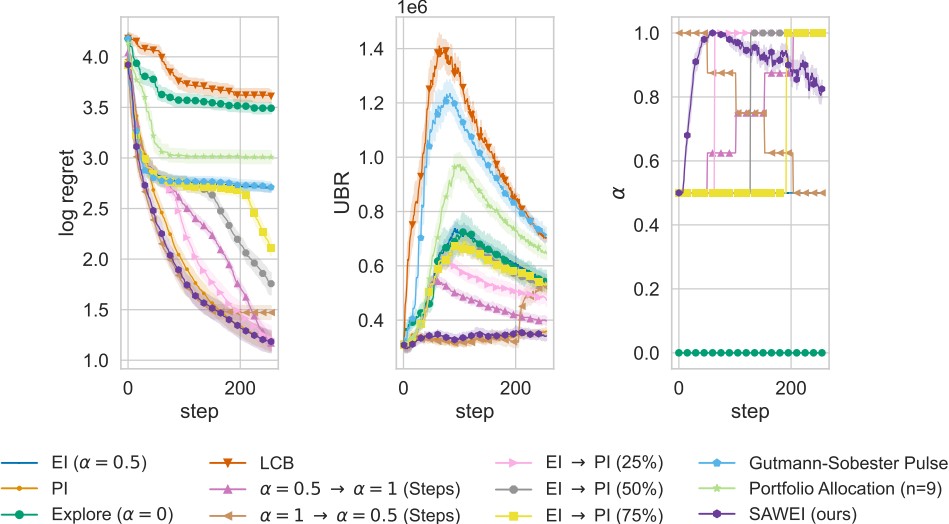

Figure 18: BBOB Function 9

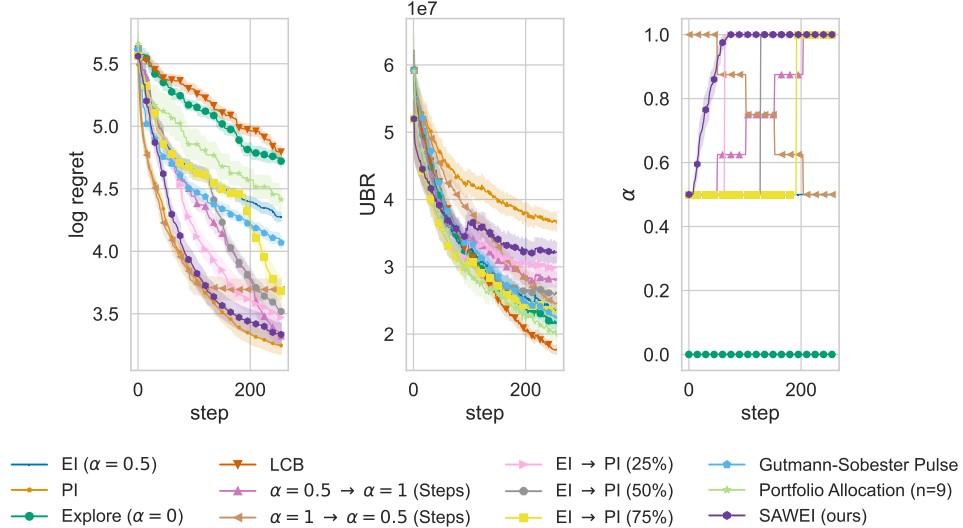

Figure 19: BBOB Function 10

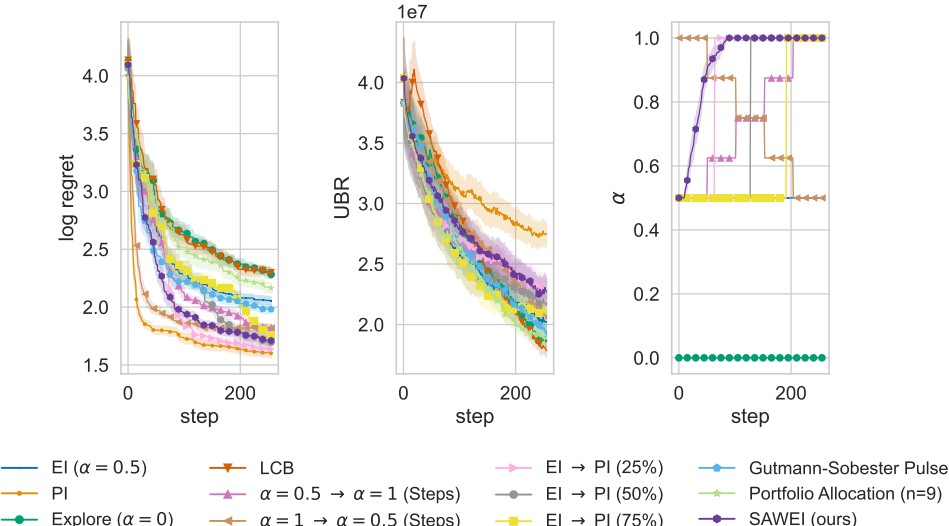

Figure 20: BBOB Function 11

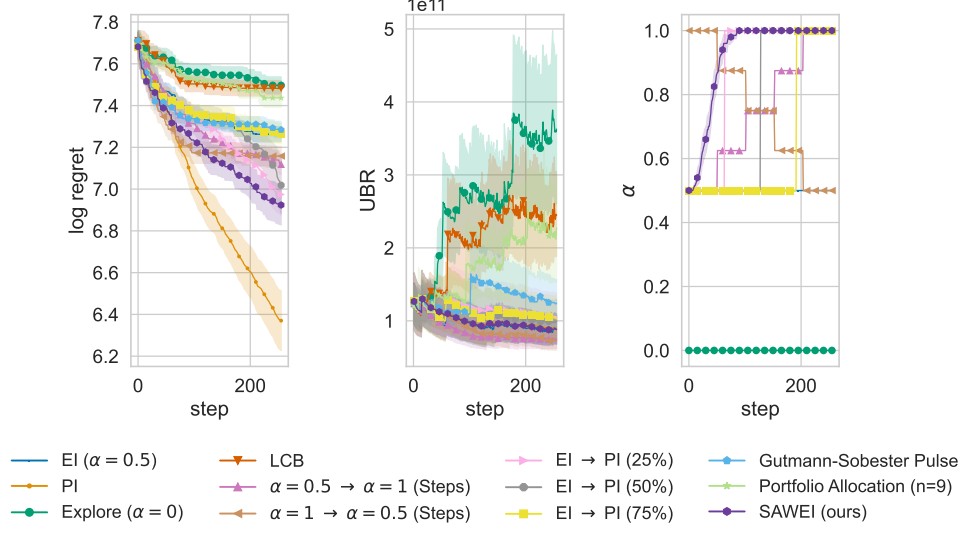

Figure 21: BBOB Function 12

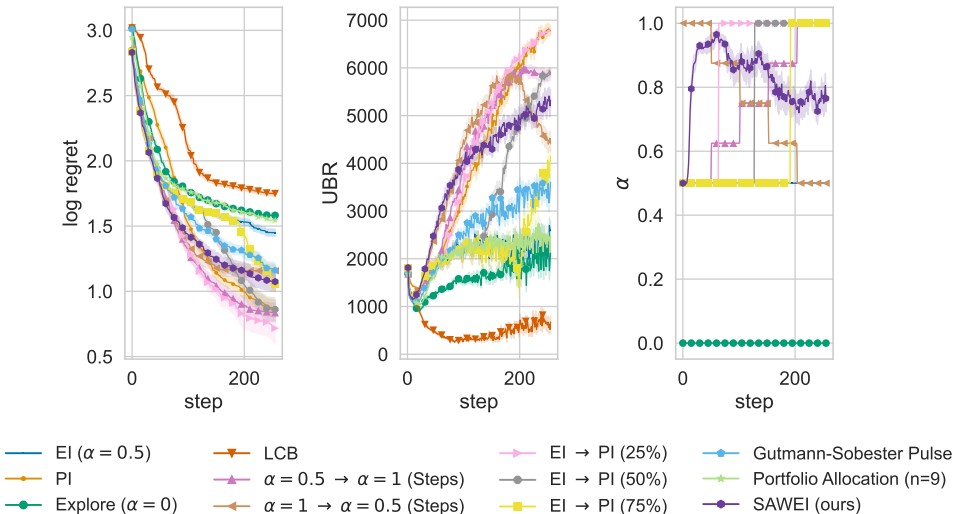

Figure 22: BBOB Function 13

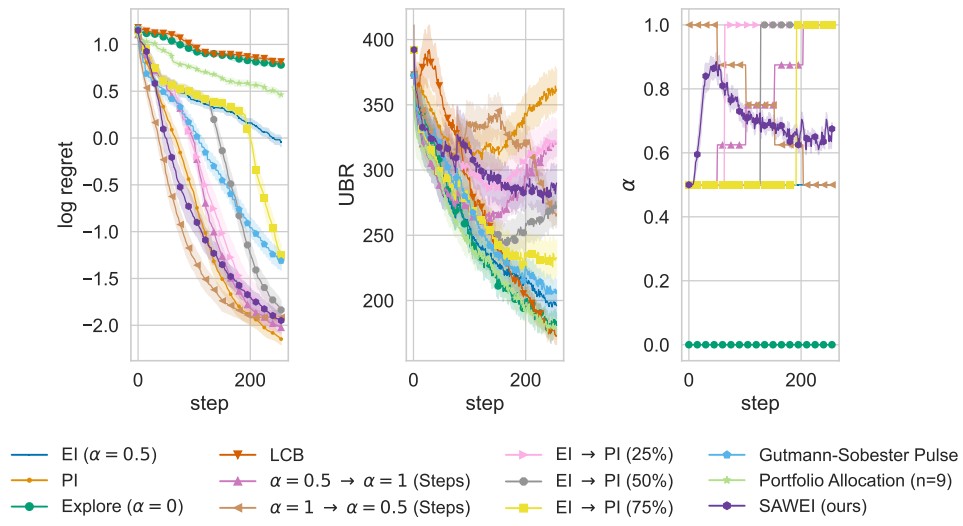

Figure 23: BBOB Function 14

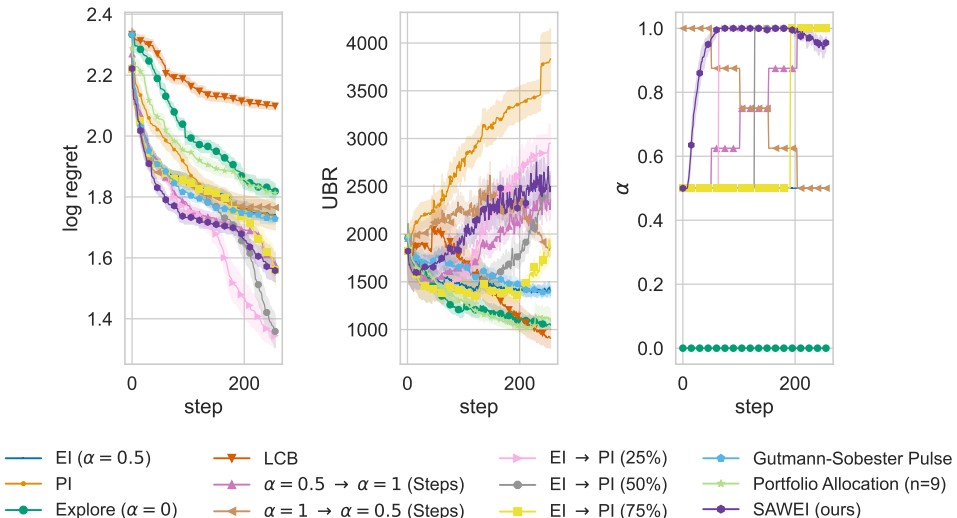

Figure 24: BBOB Function 15

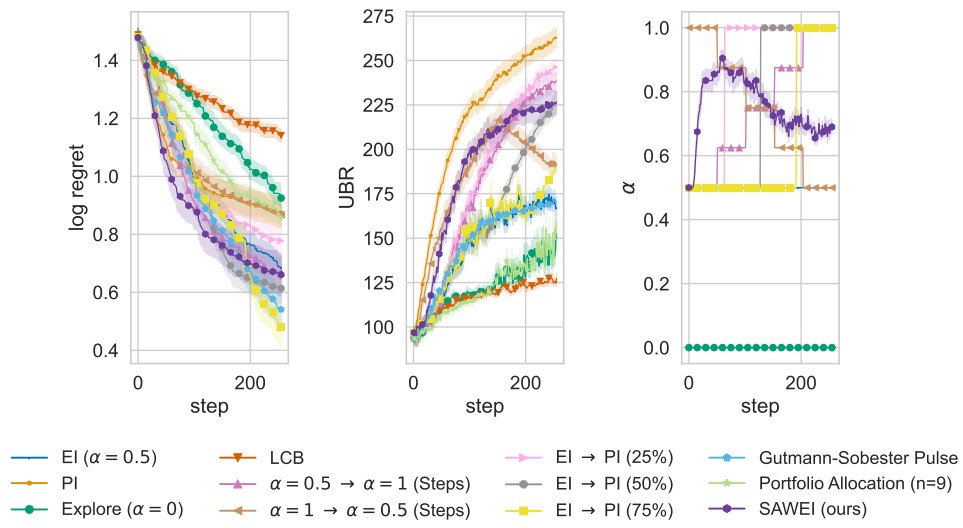

Figure 25: BBOB Function 16

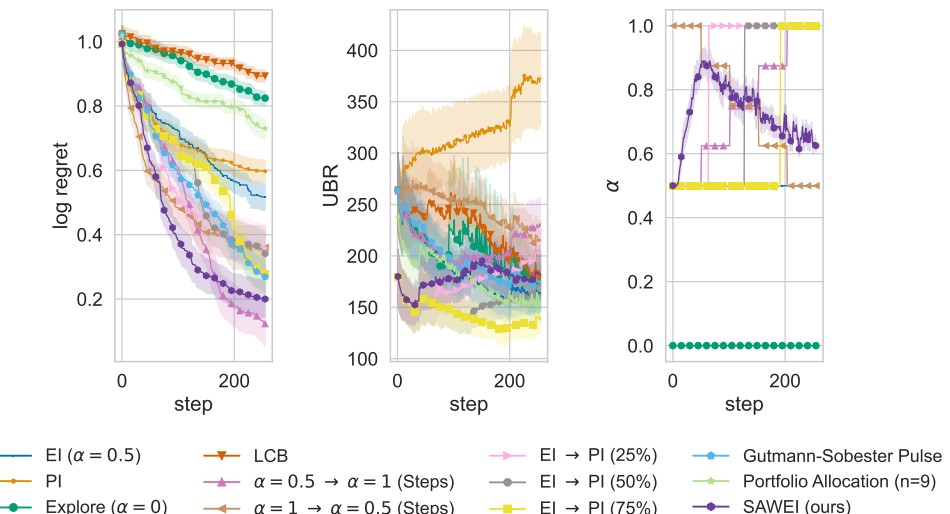

Figure 26: BBOB Function 17

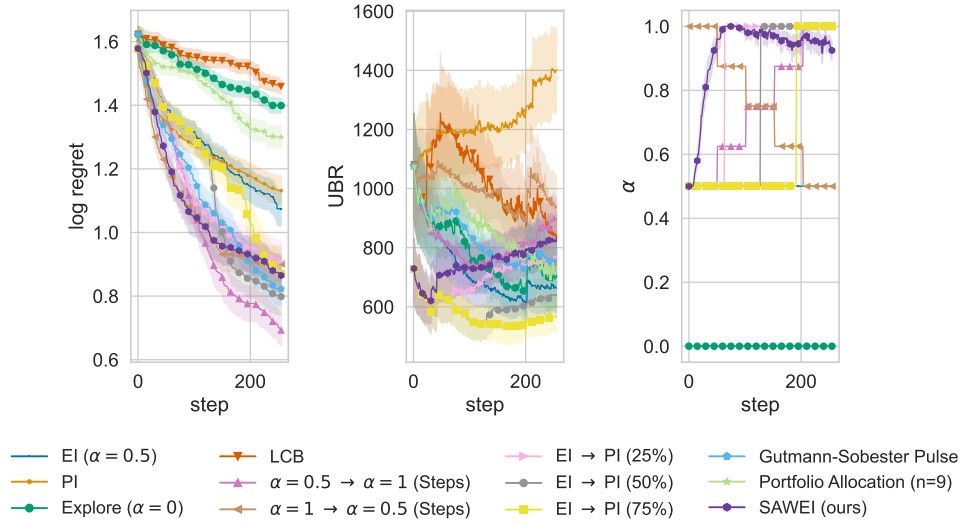

Figure 27: BBOB Function 18

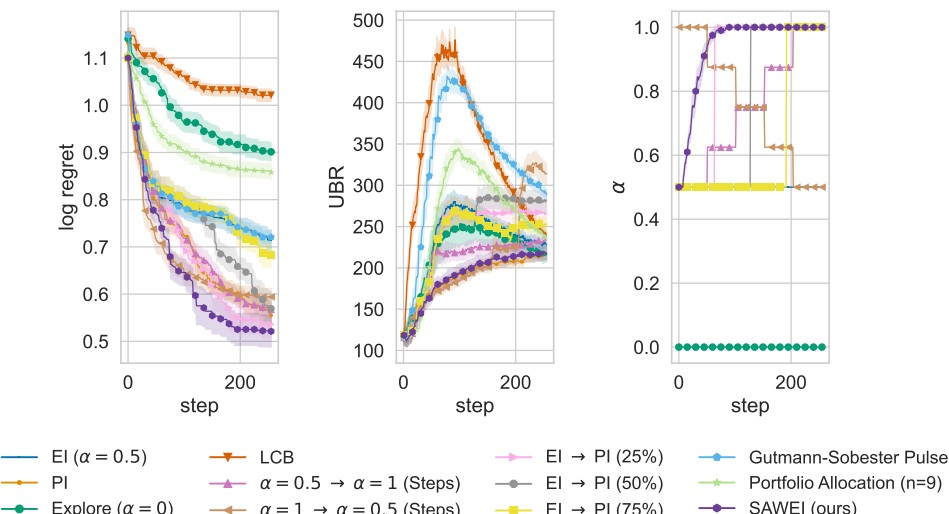

Figure 28: BBOB Function 19

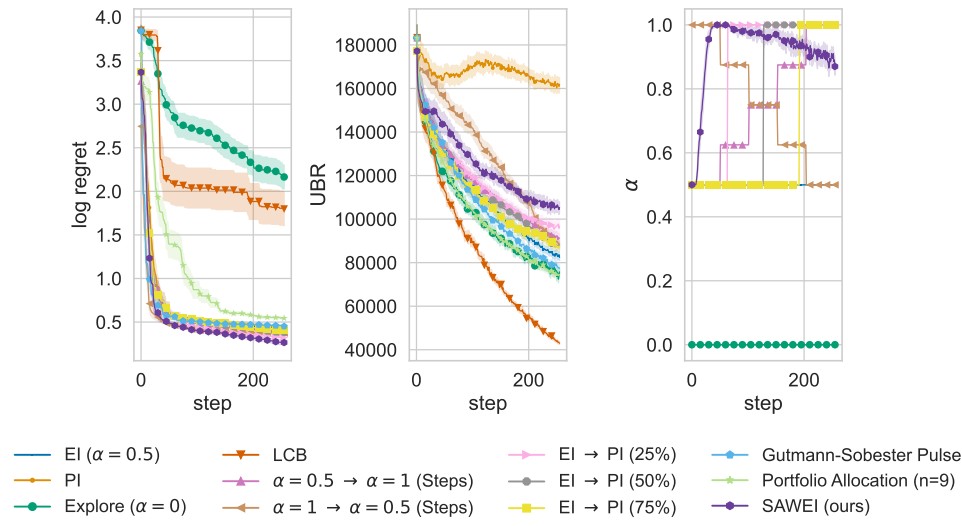

Figure 29: BBOB Function 20

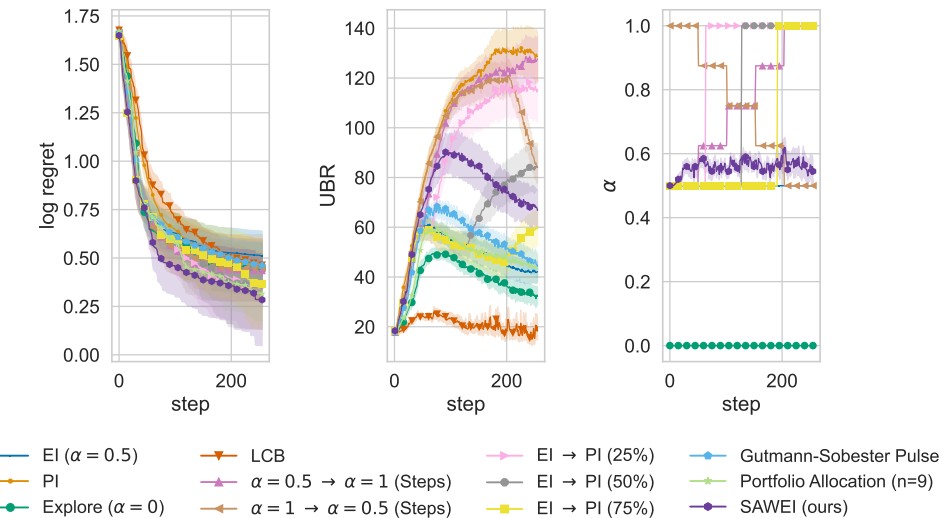

Figure 30: BBOB Function 21

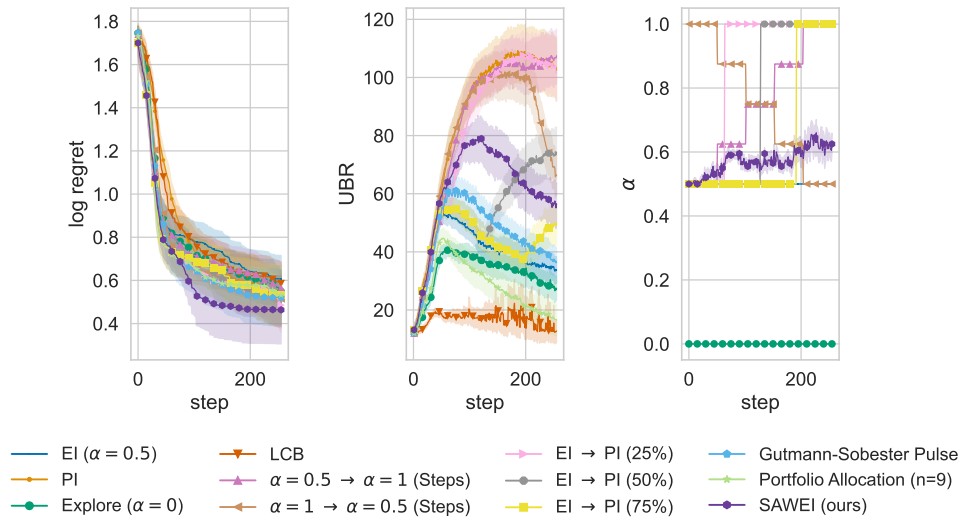

Figure 31: BBOB Function 22

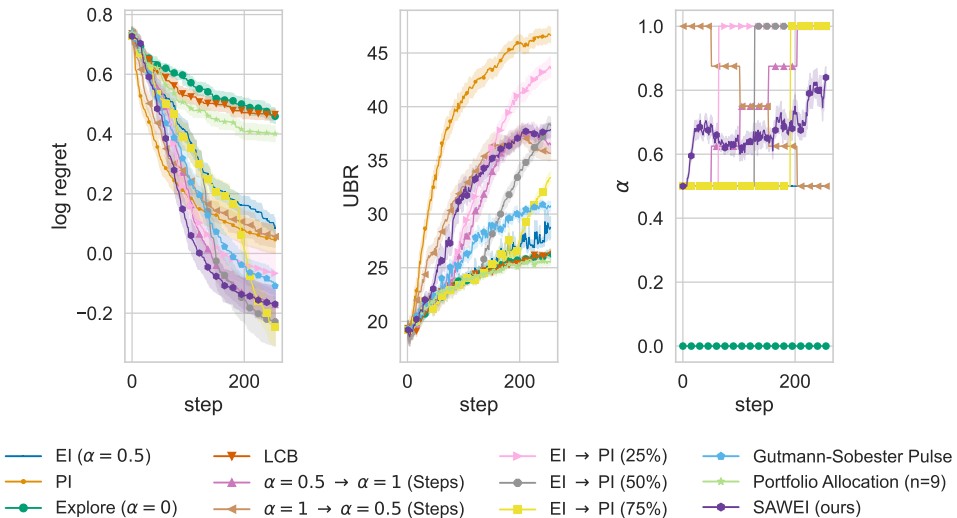

Figure 32: BBOB Function 23

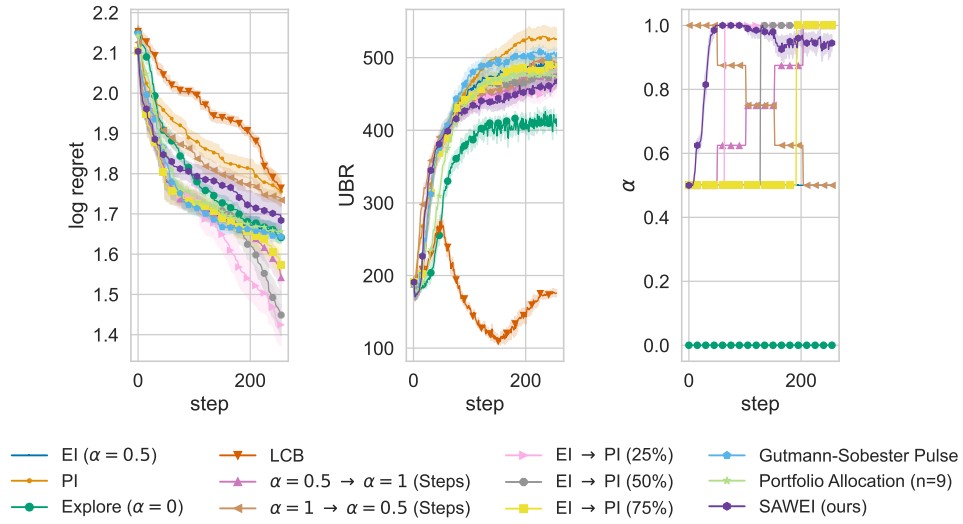

Figure 33: BBOB Function 24

# E  HPOBench Results

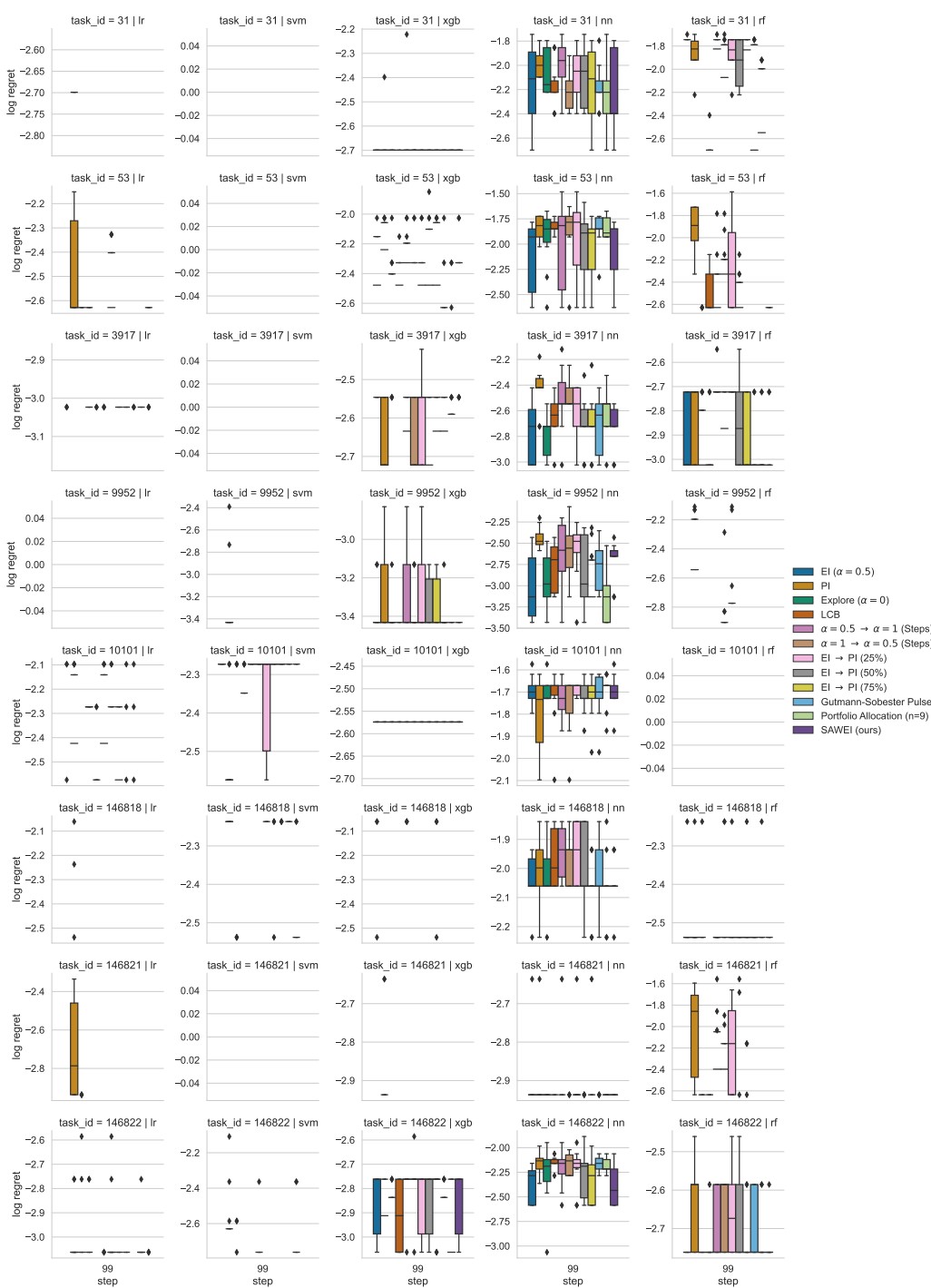

Figure 34: Final log regret on HPOBench (10 seeds). Please note that on this tabular benchmark a log regret of 0 can be achieved which is not plotted.

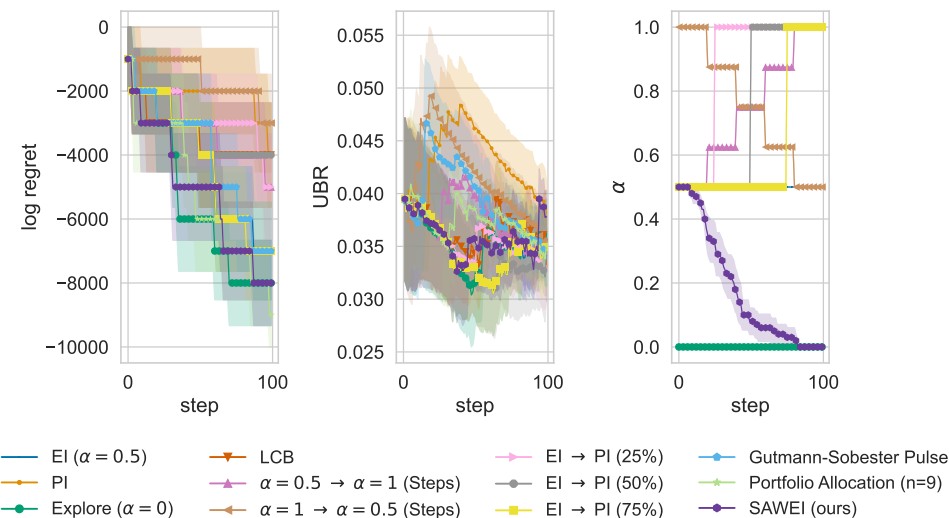

Figure 35: HPOBench ML: (model, task_id) = (lr, 10101)

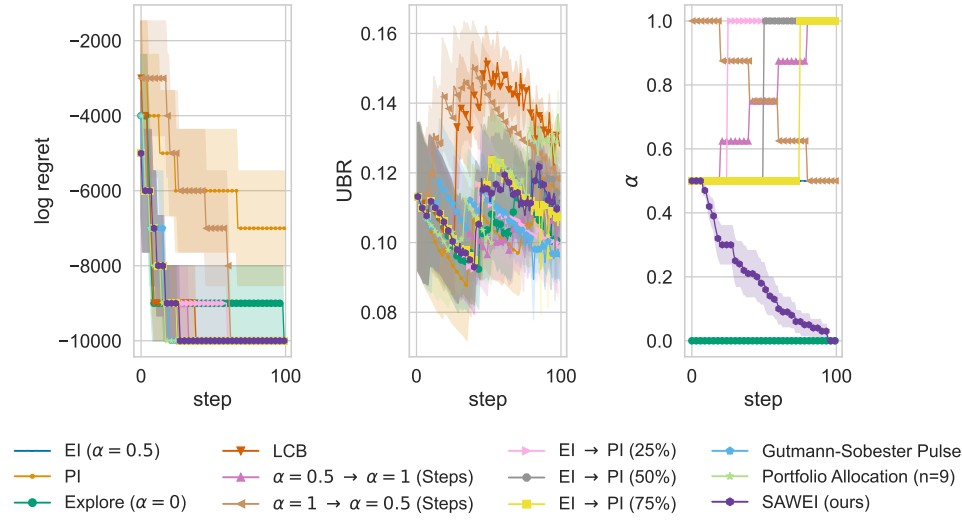

Figure 36: HPOBench ML: (model, task_id) = (lr, 146818)

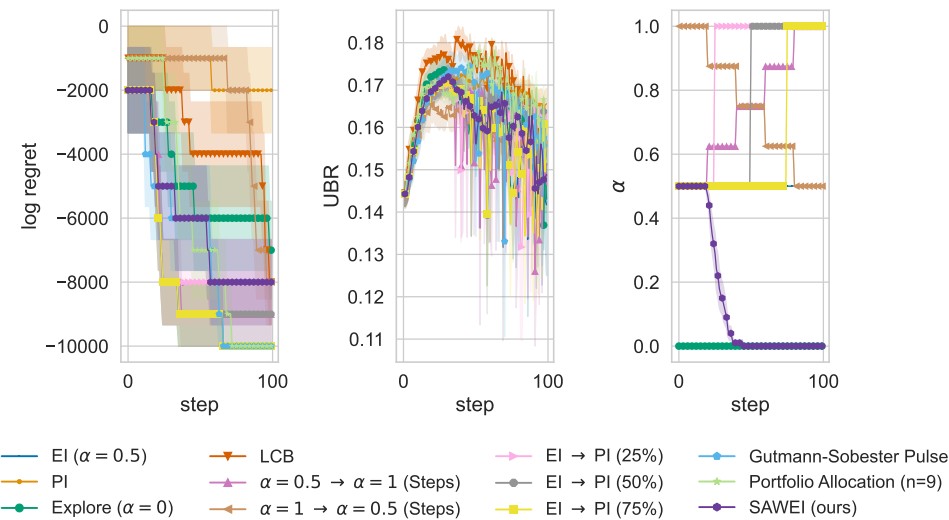

Figure 37: HPOBench ML: (model, task_id) = (lr, 146821)

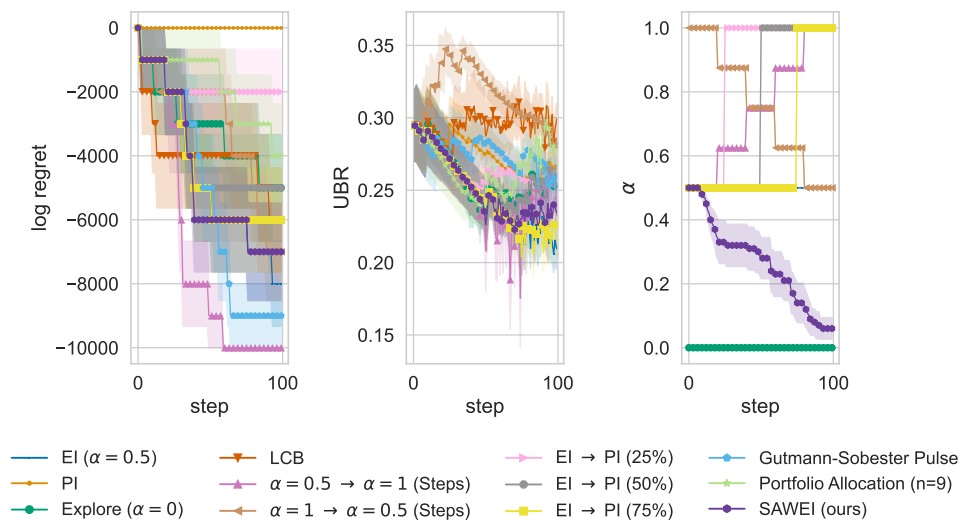

Figure 38: HPOBench ML: (model, task_id) = (lr, 146822)

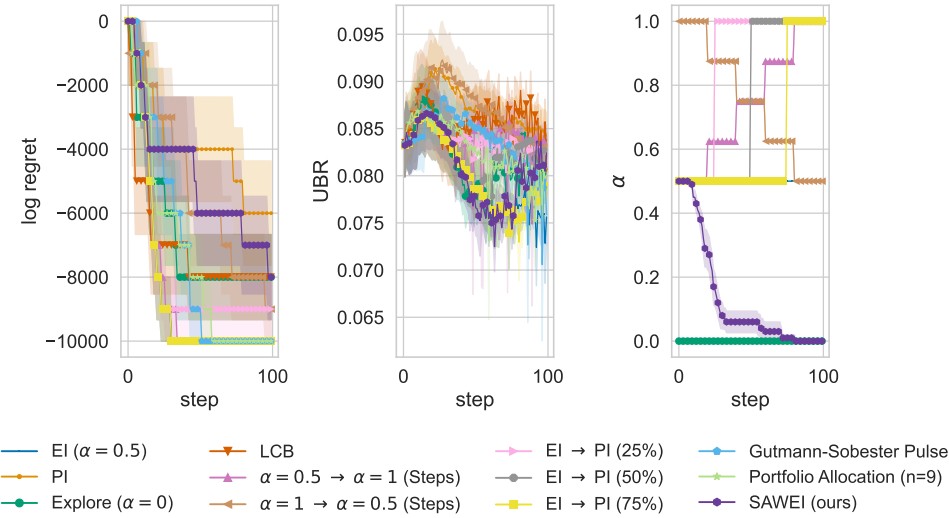

Figure 39: HPOBench ML: (model, task_id) = (lr, 31)

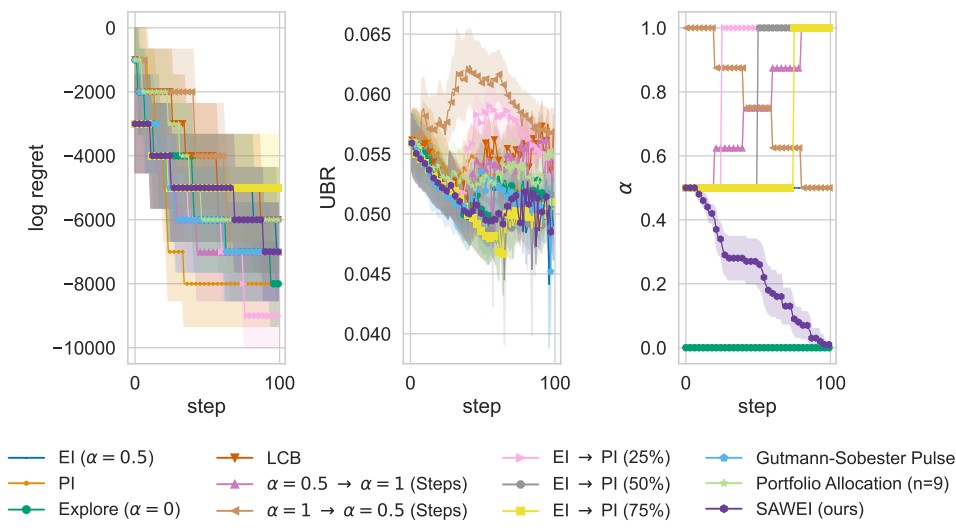

Figure 40: HPOBench ML: (model, task_id) = (lr, 3917)

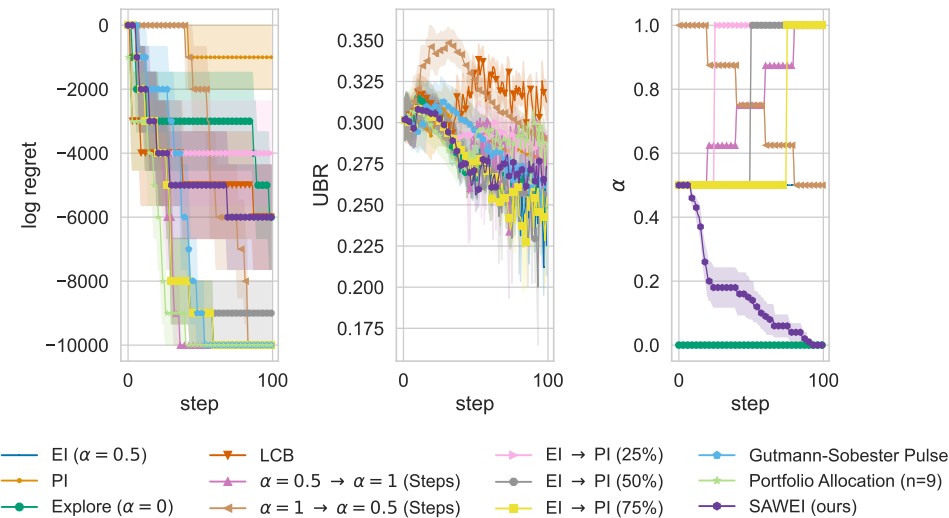

Figure 41: HPOBench ML: (model, task_id) = (lr, 53)

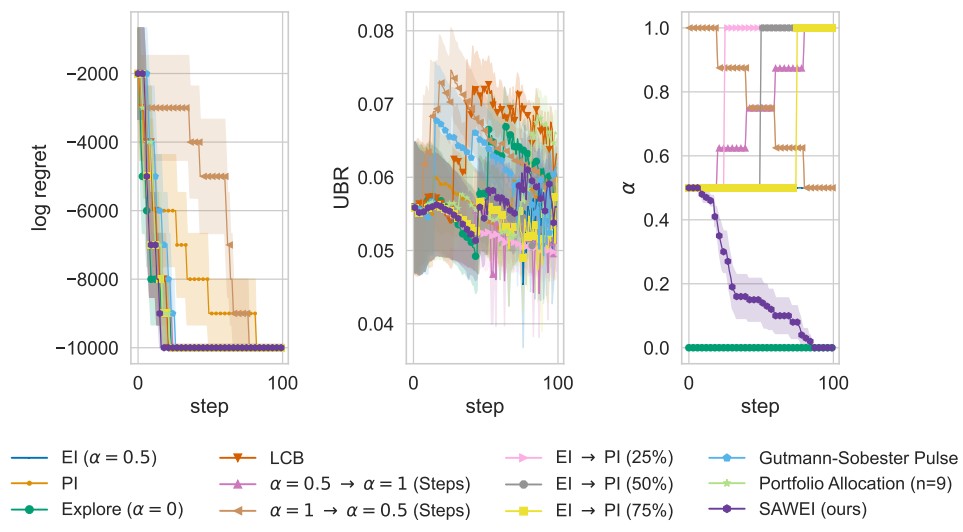

Figure 42: HPOBench ML: (model, task_id) = (lr, 9952)

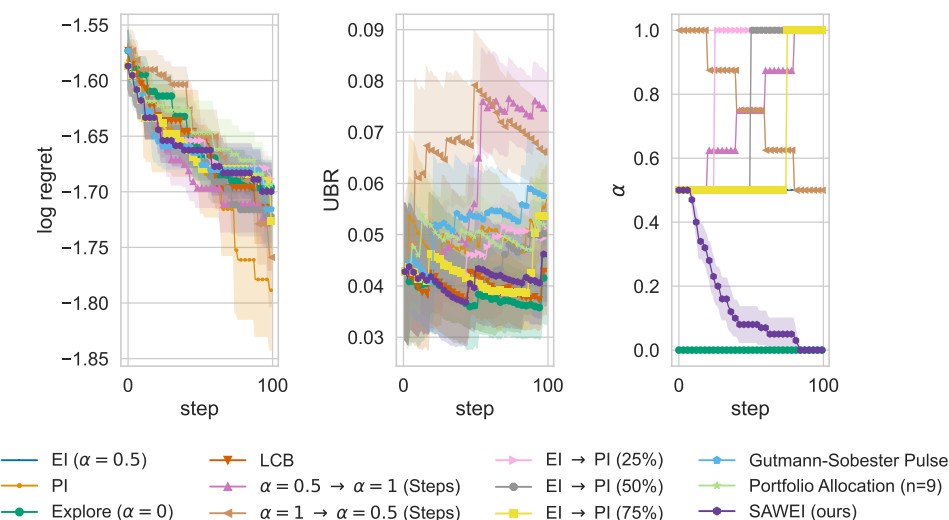

Figure 43: HPOBench ML: (model, task_id) = (nn, 10101)

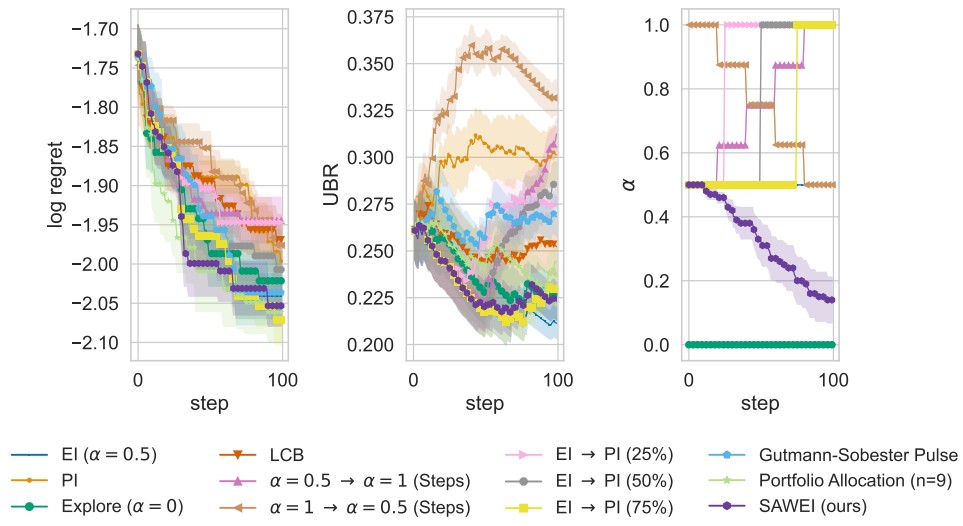

Figure 44: HPOBench ML: (model, task_id) = (nn, 146818)

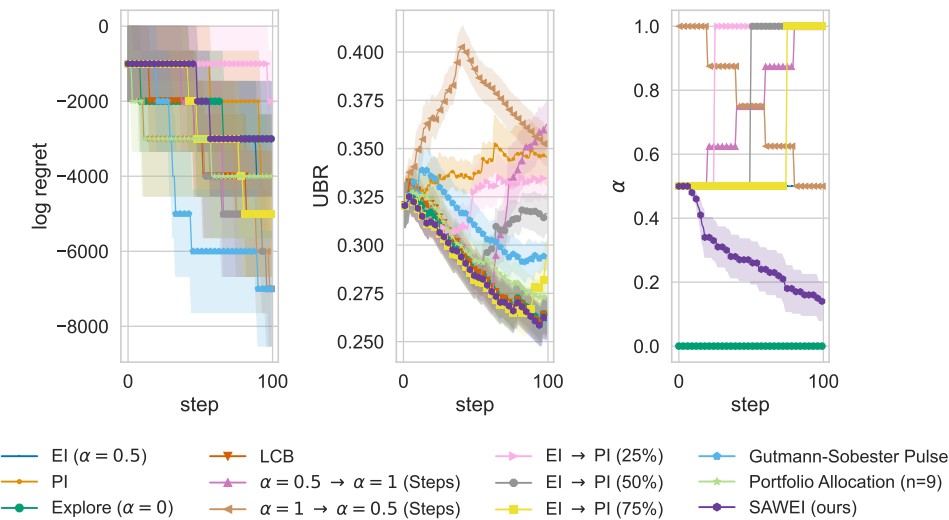

Figure 45: HPOBench ML: (model, task_id) = (nn, 146821)

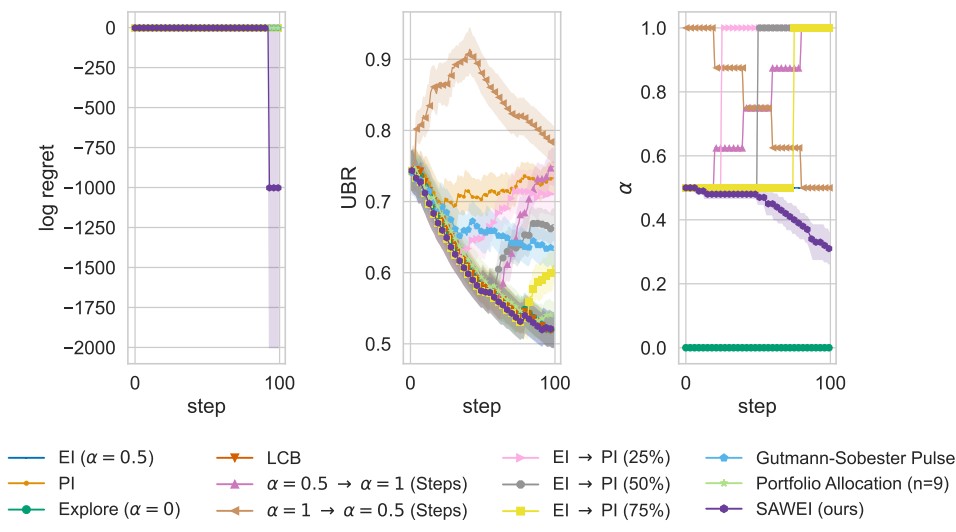

Figure 46: HPOBench ML: (model, task_id) = (nn, 146822)

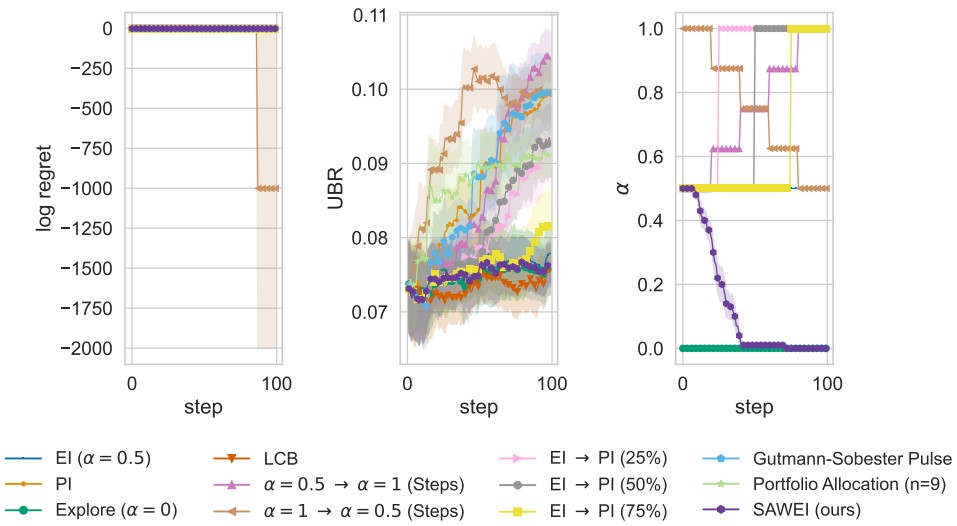

Figure 47: HPOBench ML: (model, task_id) = (nn, 31)

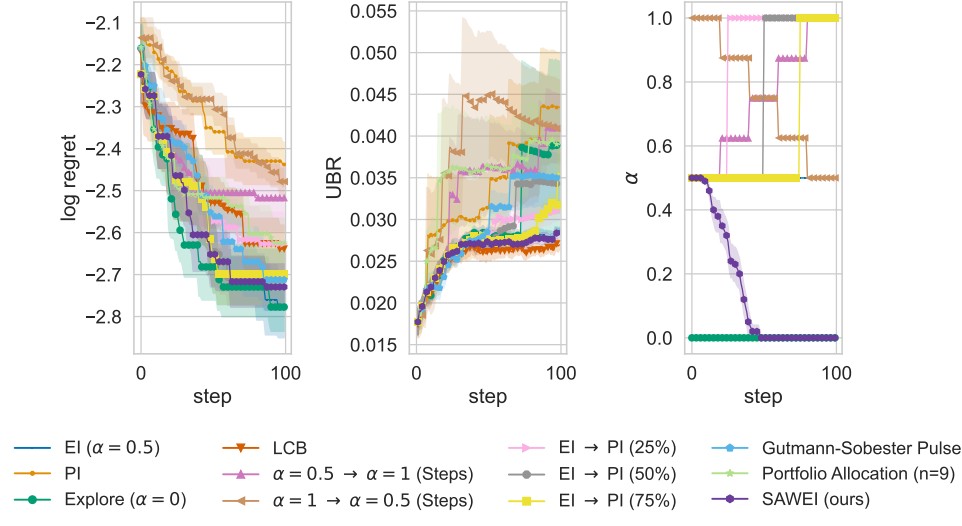

Figure 48: HPOBench ML: (model, task_id) = (nn, 3917)

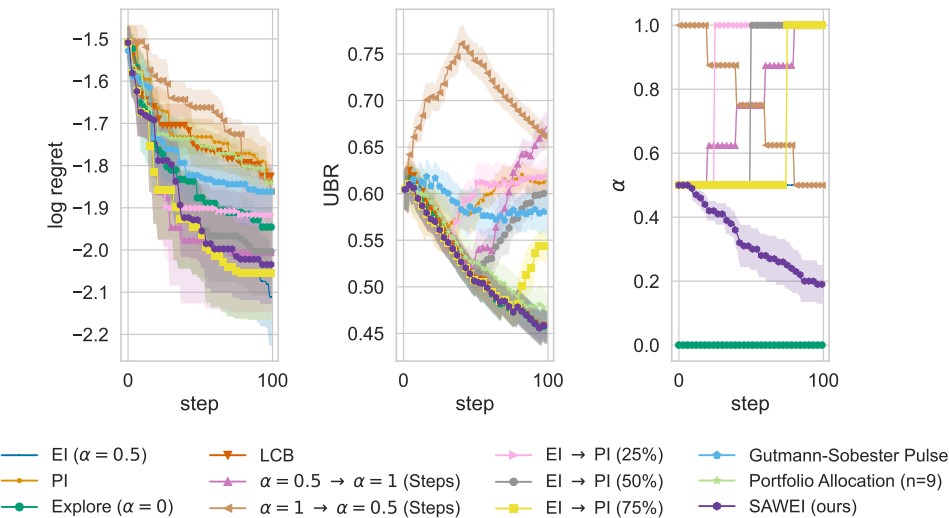

Figure 49: HPOBench ML: (model, task_id) = (nn, 53)

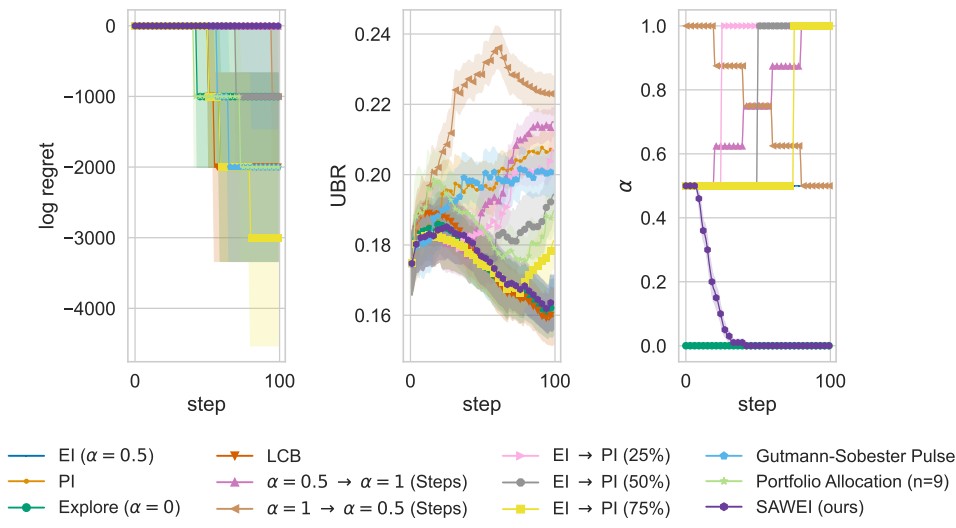

Figure 50: HPOBench ML: (model, task_id) = (nn, 9952)

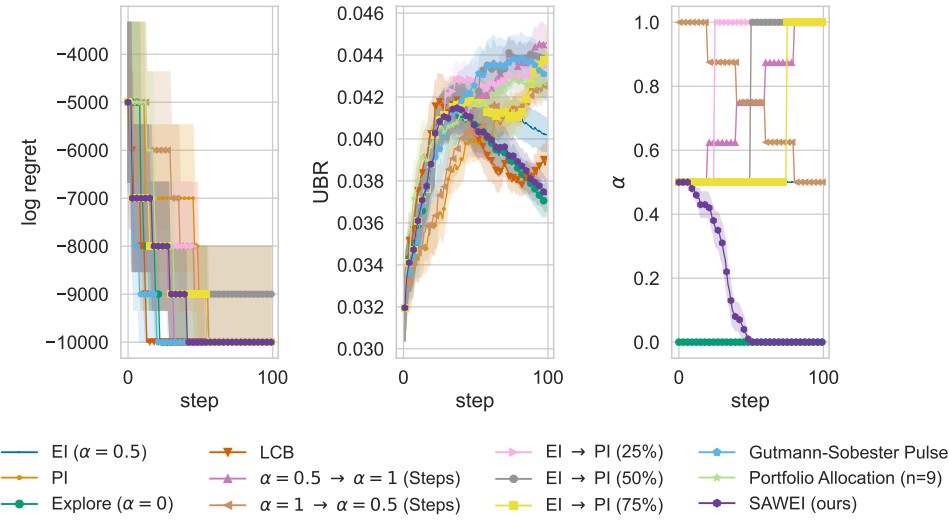

Figure 51: HPOBench ML: (model, task_id) = (rf, 10101)

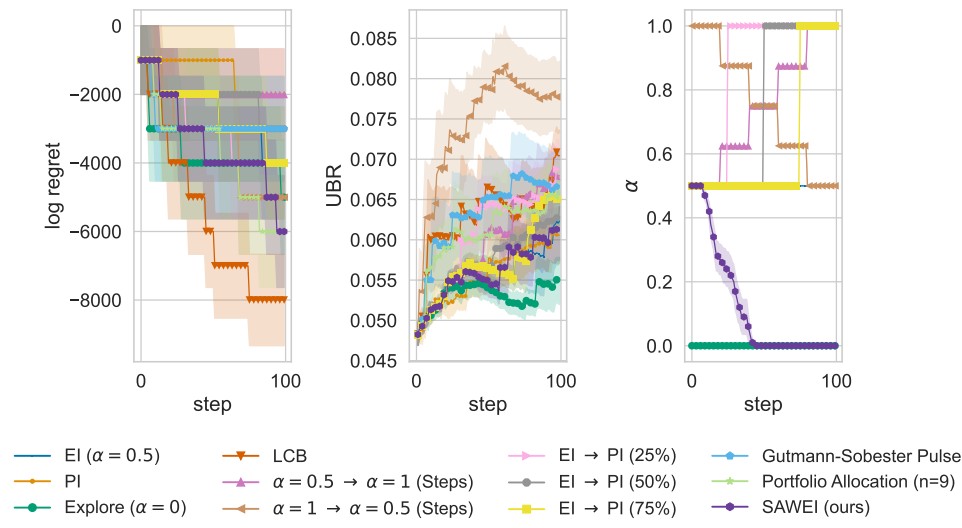

Figure 52: HPOBench ML: (model, task_id) = (rf, 146818)

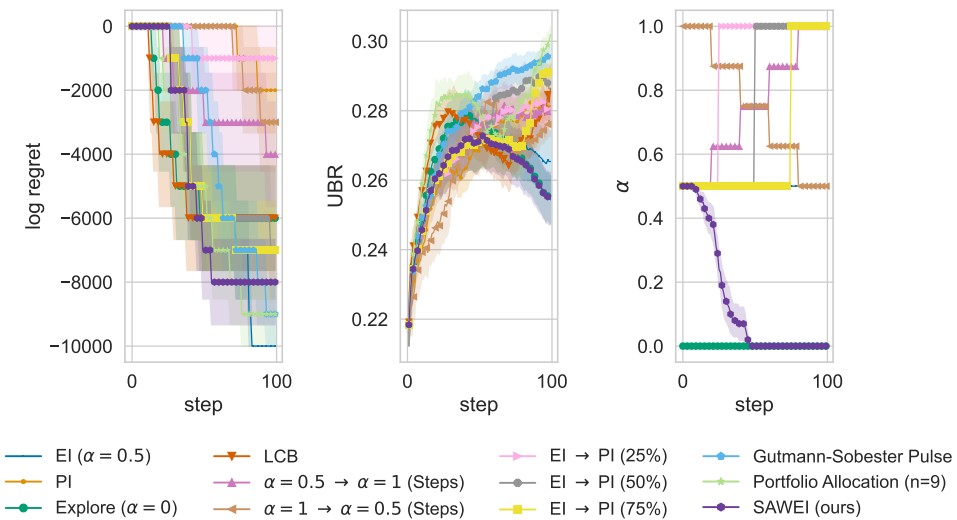

Figure 53: HPOBench ML: (model, task_id) = (rf, 146821)

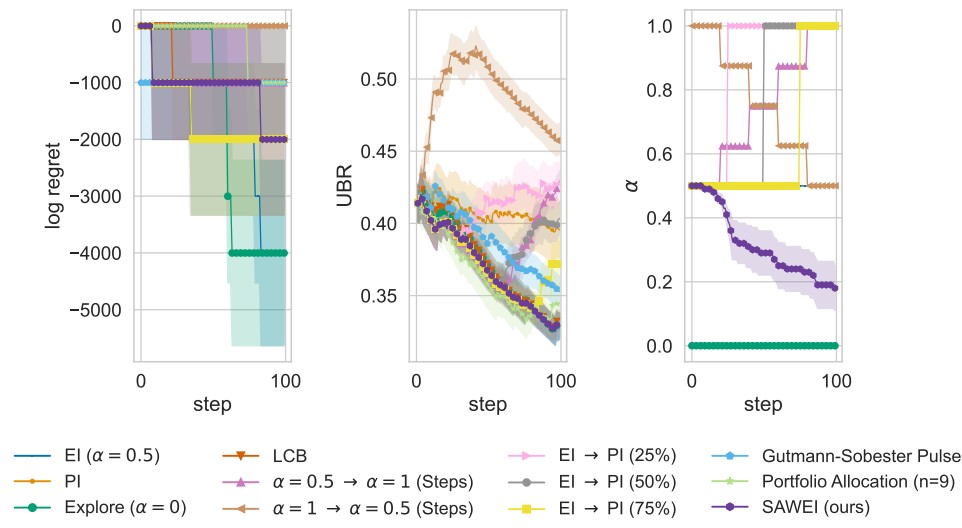

Figure 54: HPOBench ML: (model, task_id) = (rf, 146822)

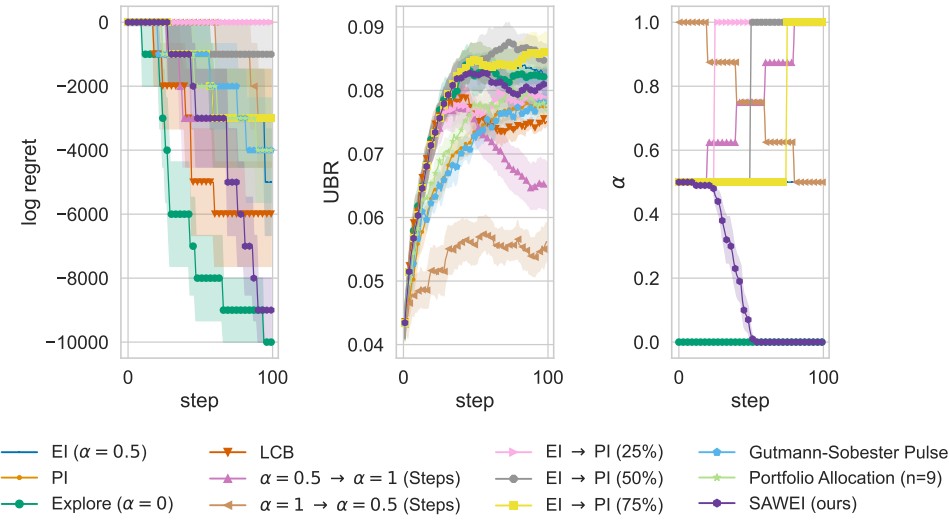

Figure 55: HPOBench ML: (model, task_id) = (rf, 31)

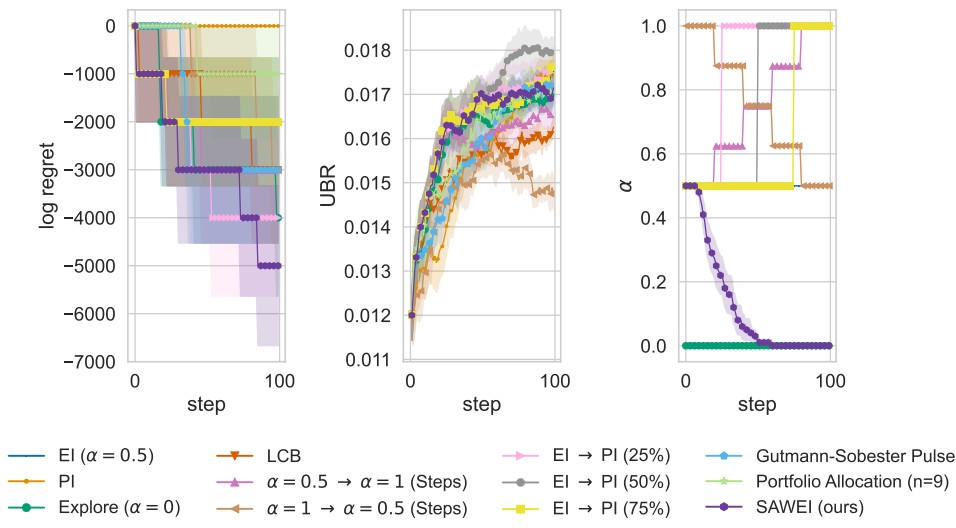

Figure 56: HPOBench ML: (model, task_id) = (rf, 3917)

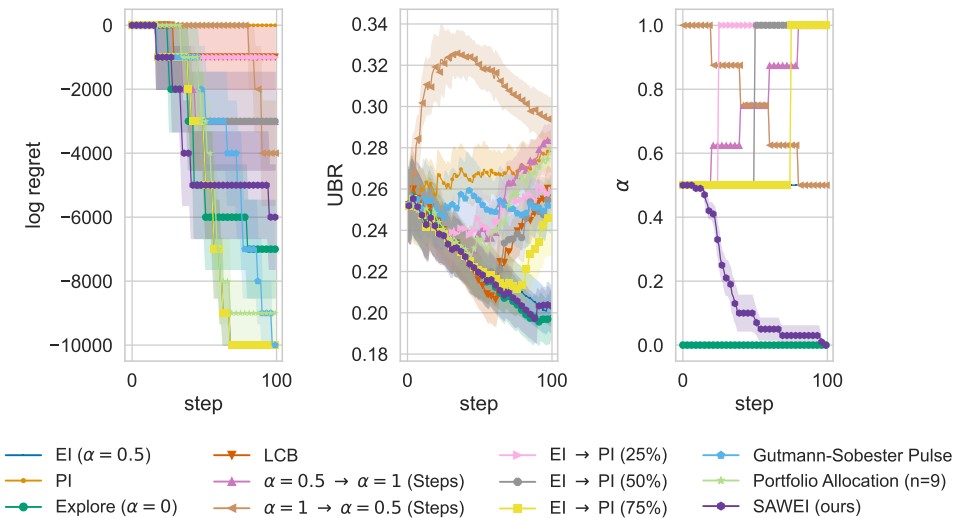

Figure 57: HPOBench ML: (model, task_id) = (rf, 53)

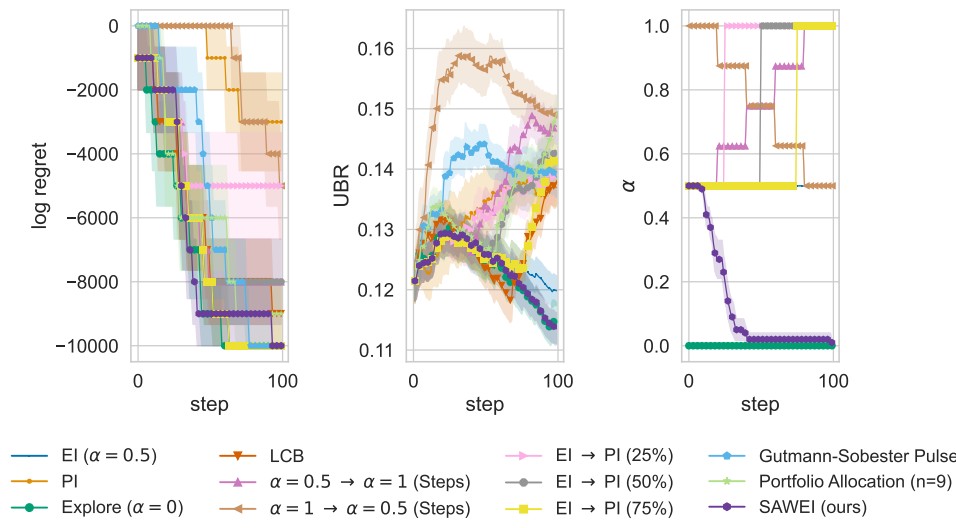

Figure 58: HPOBench ML: (model, task_id) = (rf, 9952)

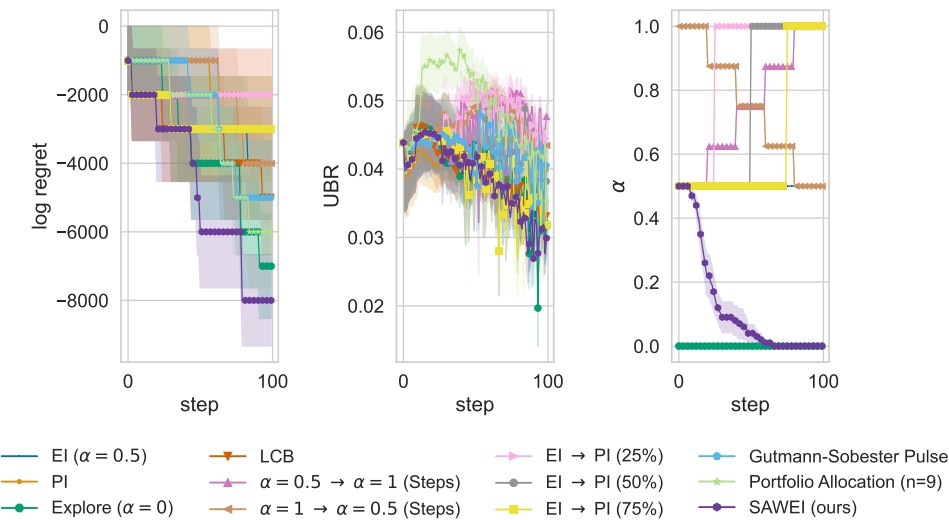

Figure 59: HPOBench ML: (model, task_id) = (svm, 10101)

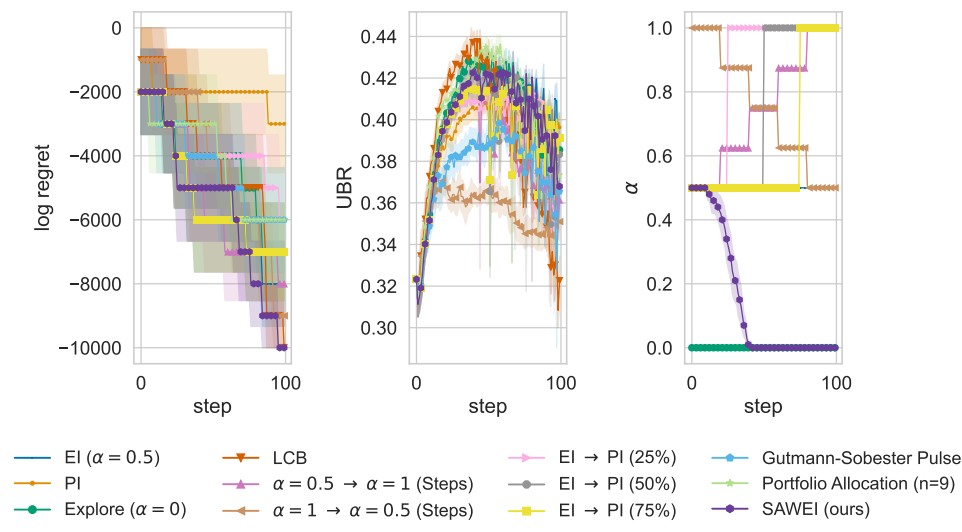

Figure 60: HPOBench ML: (model, task_id) = (svm, 146818)

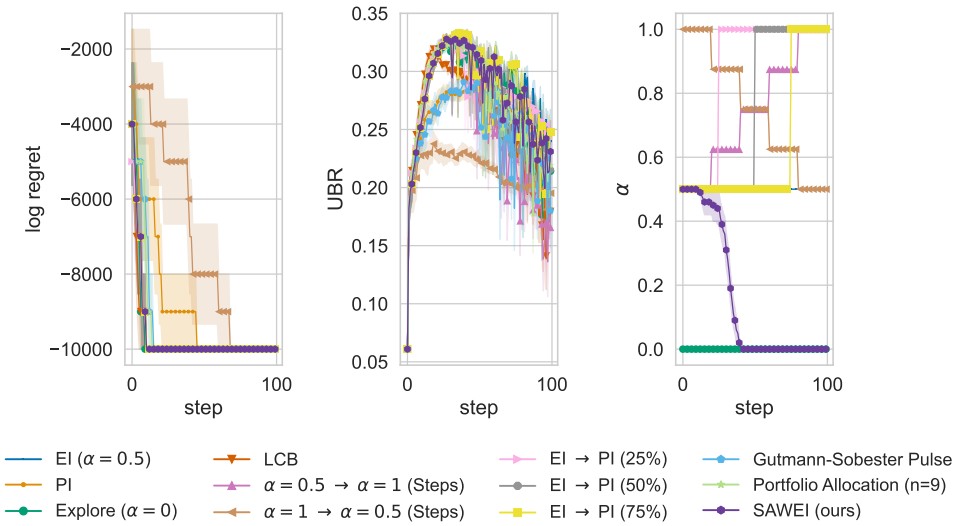

Figure 61: HPOBench ML: (model, task_id) = (svm, 146821)

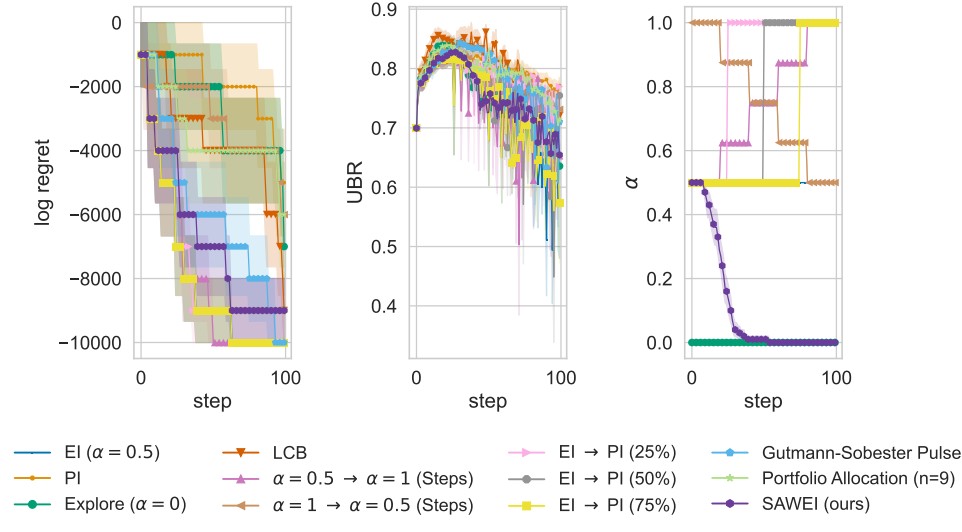

Figure 62: HPOBench ML: (model, task_id) = (svm, 146822)

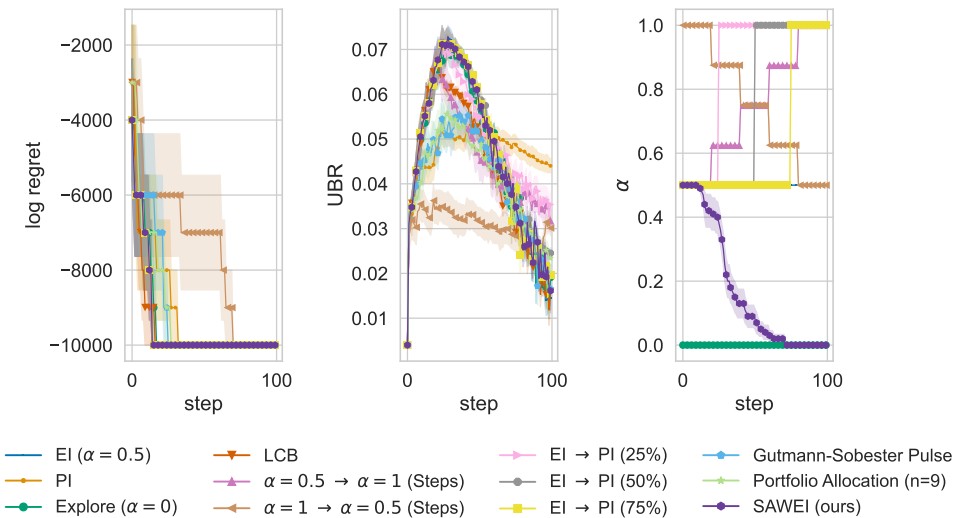

Figure 63: HPOBench ML: (model, task_id) = (svm, 31)

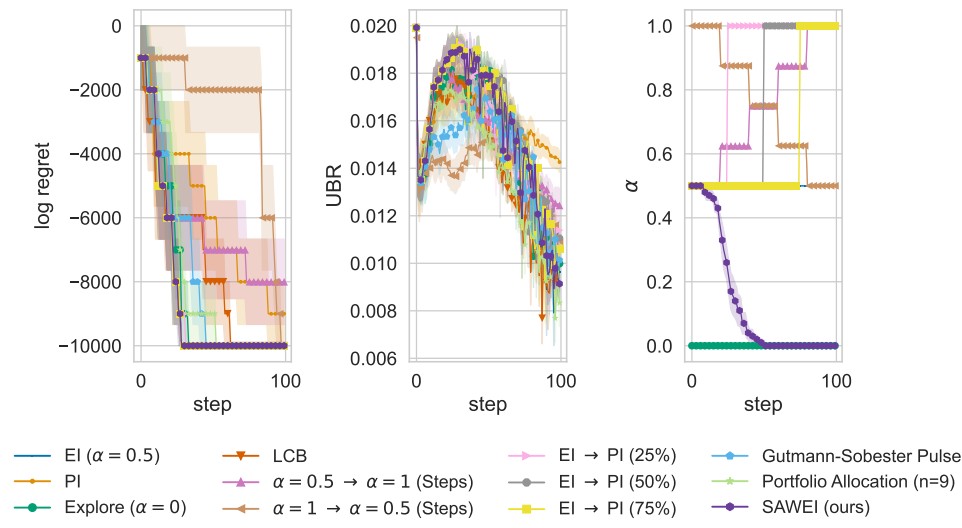

Figure 64: HPOBench ML: (model, task_id) = (svm, 3917)

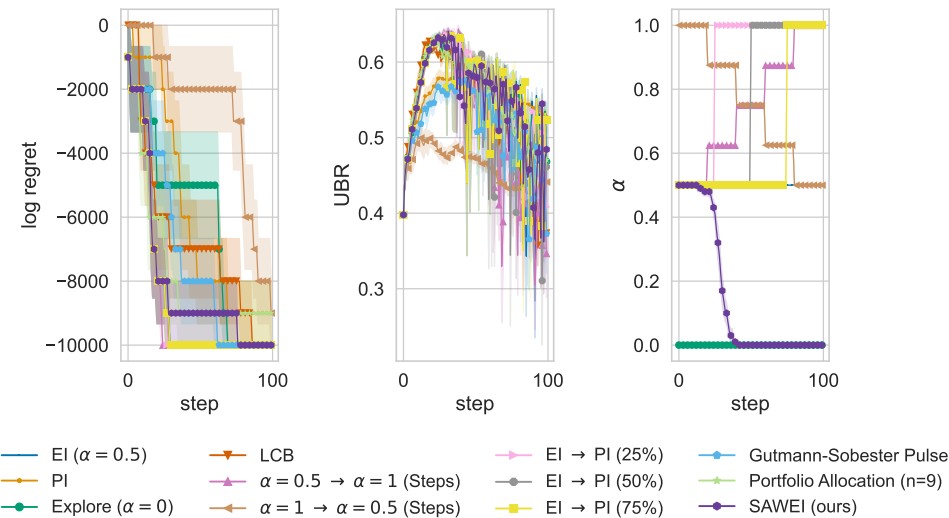

Figure 65: HPOBench ML: (model, task_id) = (svm, 53)

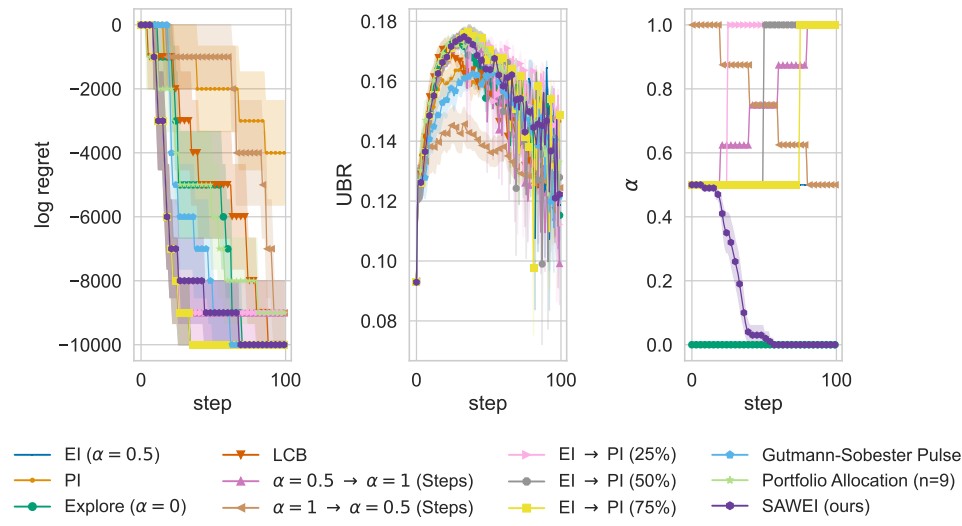

Figure 66: HPOBench ML: (model, task_id) = (svm, 9952)

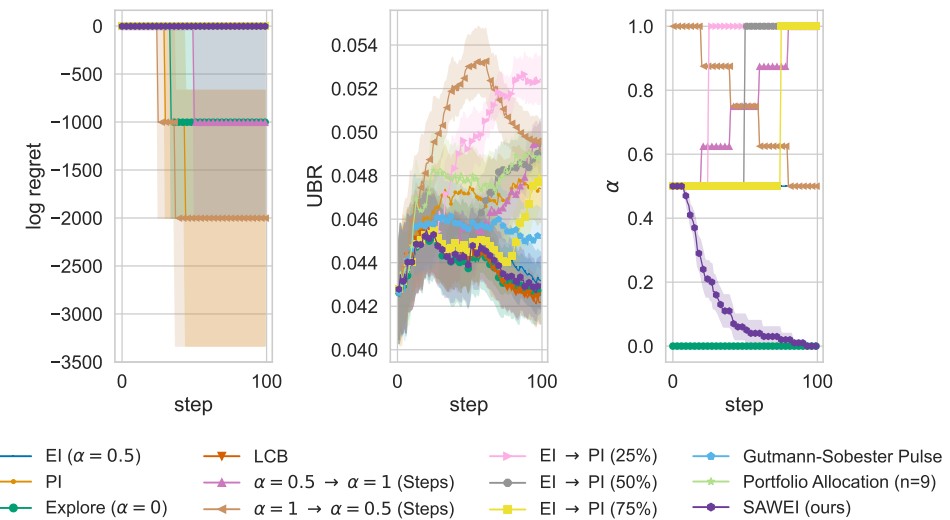

Figure 67: HPOBench ML: (model, task_id) = (xgb, 10101)

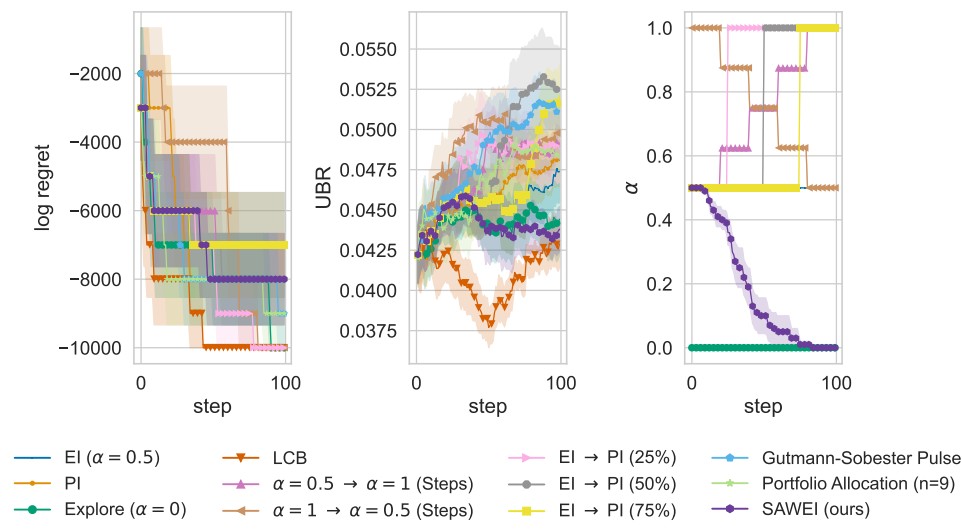

Figure 68: HPOBench ML: (model, task_id) = (xgb, 146818)

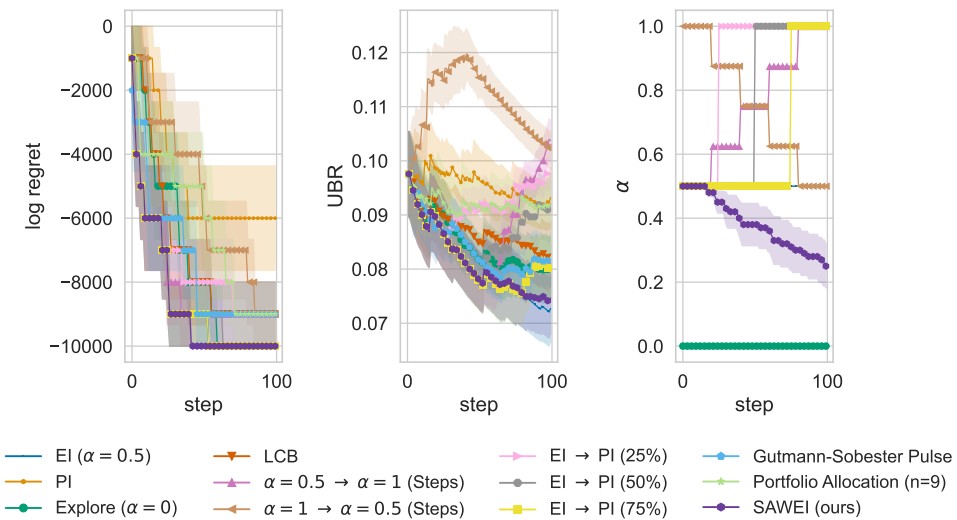

Figure 69: HPOBench ML: (model, task_id) = (xgb, 146821)

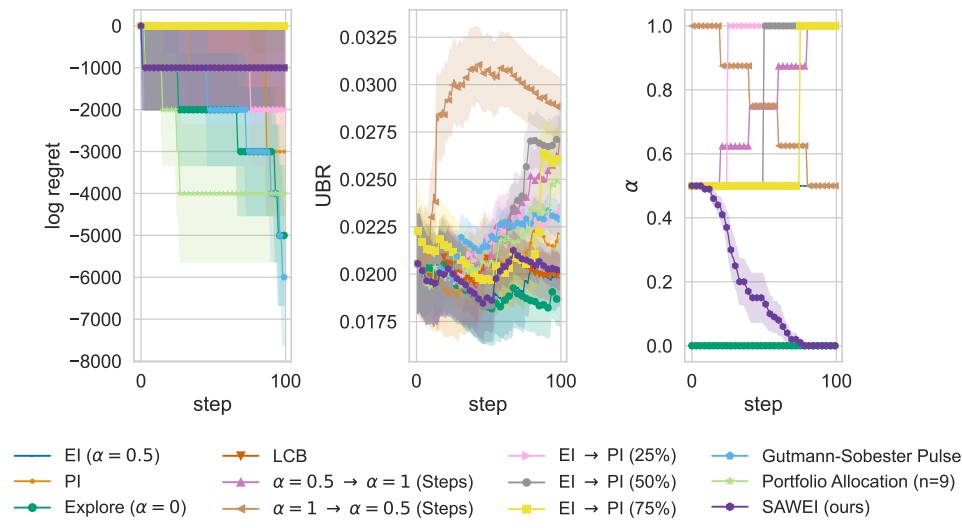

Figure 70: HPOBench ML: (model, task_id) = (xgb, 146822)

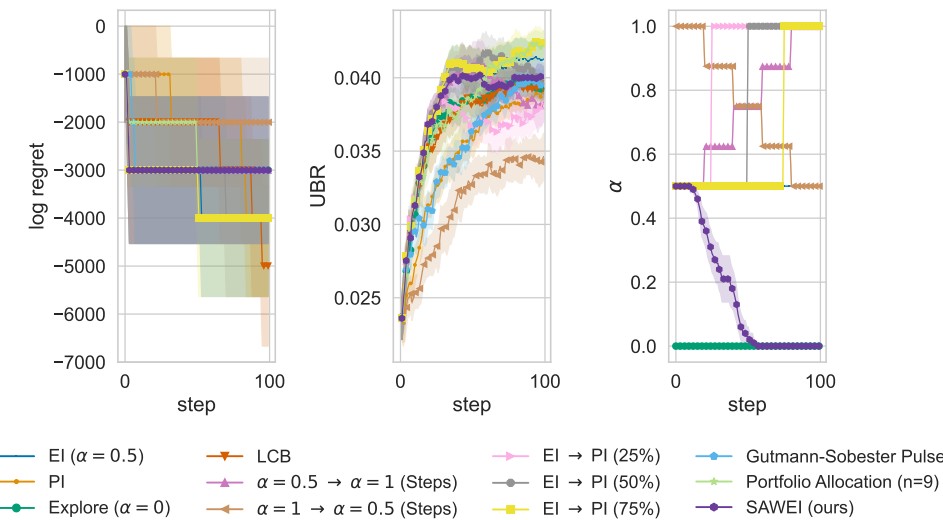

Figure 71: HPOBench ML: (model, task_id) = (xgb, 31)

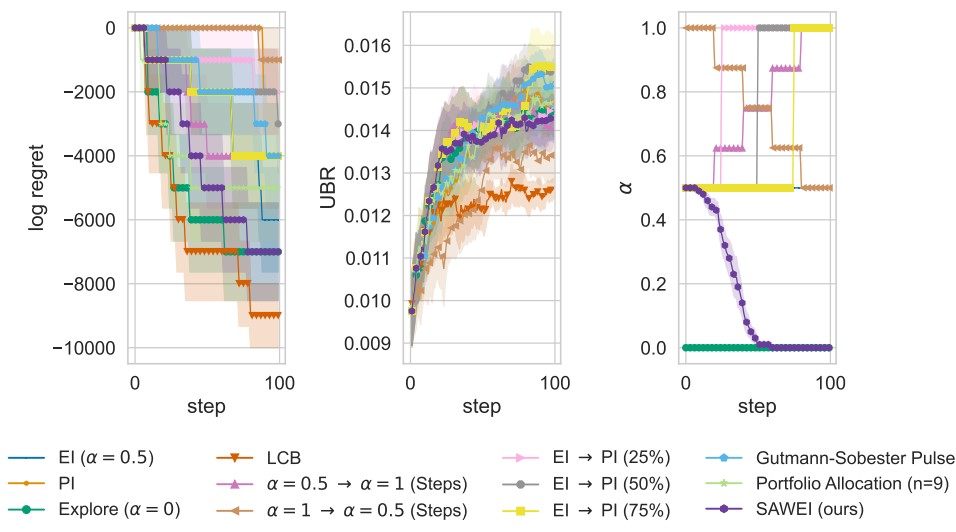

Figure 72: HPOBench ML: (model, task_id) = (xgb, 3917)

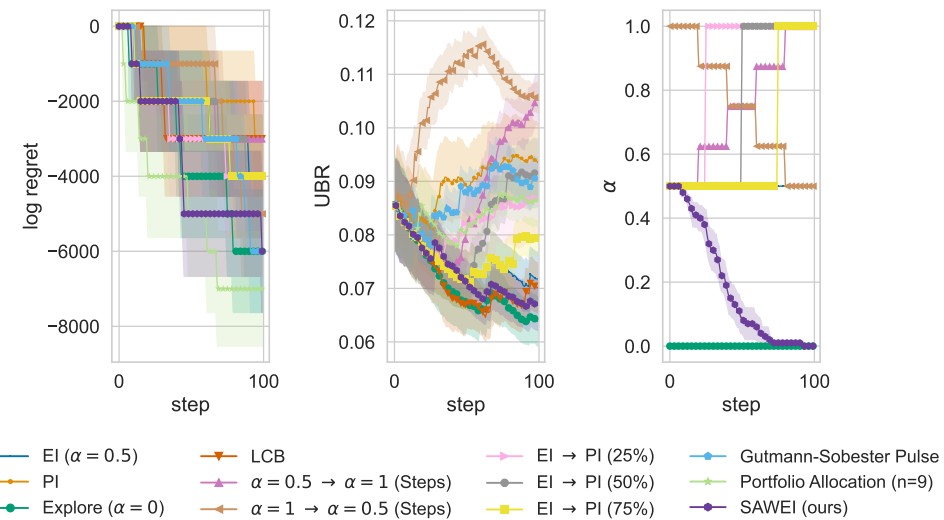

Figure 73: HPOBench ML: (model, task_id) = (xgb, 53)

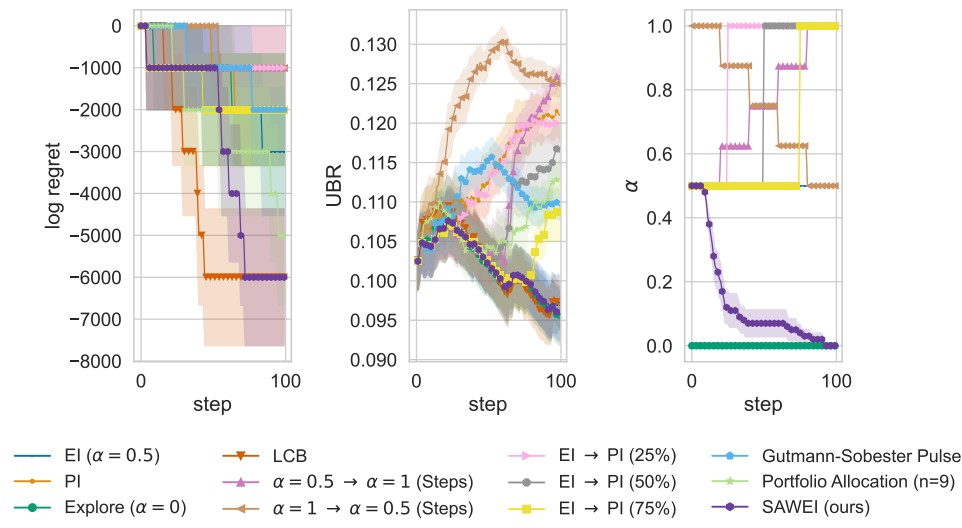

Figure 74: HPOBench ML: (model, task_id) = (xgb, 9952)

