# OpenReview forum: "Self-Adjusting Weighted Expected Improvement for Bayesian Optimization"
_automl.cc/AutoML/2023/Conference — AutoML 2023 MainTrack_

### Review · Reproducibility_Reviewer_CqGE · 2023-03-31

**Completeness Of Code And Dataset Supplement Rating:** 3
**Usability And Ease Of Reproducibility Rating:** 4

**Actions Required To Increase The Reproducibility And Overall Recommendation:**

I would like to see the following improvements:
* fix typos that prevent automatic install
* provide more self-contained and explained examples that do not require a slurm setup
* provide the raw benchmark results to be able to exactly reproduce the figures and results as reported in the paper
* fix issues that currently prevent easy running of code examples


**Completeness Of Code And Dataset Supplement:**

Code appears to be complete, except for raw results of the original benchmark runs on BBOB and HPOBench not being provided.
It would have been nice to provide these data to be able to reproduce the figures and results reported in the paper.
Also, it would have been nice to directly state which code snippets can be used to reproduce which figure.

**Overall Reproducibility Review:**

Overall I believe that all code is provided that is required to reproduce the results and figures reported in the paper.
While the setup itself currently requires slightly more effort than a single `pip install -e .`, general documentation regarding the setup process is good.
On the other hand, minimal examples and their documentation could be improved.
Currently, it very much feels like a code dump without that much effort being put into usability.

**Review Confidence:**

4: You are confident in your assessment, but not absolutely certain. It is unlikely, but not impossible, that you did not understand some parts of the submission or that you are unfamiliar with some pieces of the code or data.

**Review Rating:**

8: Accept, all aspects of this are reproducible with minor effort.

**Review Summary:**

All necessary files required to rerun benchmark experiments and reproduce results and figures are available.
Rerunning experiments currently fails due to some issues with the hydra setup (?) but I believe this is easily fixable.
All in all these things can be easily fixed by the authors and I am very much willing to increase my rating during the rebuttal as I believe that the code submission itself is of good quality.

**Summary Of Necessary Code And Dataset Supplement:**

The paper proposes a self-adjusting scheduler for the Weighted Expected Improvement for BO (SAWEI) and compares it to other BO variants. Main computational aspects include: 1) A BO loop (the authors rely on SMAC) 2) Implementation of SAWEI 3) Implementation of baselines (again relying on SMAC) 4) Benchmarking on BBOB and HPOBench 5) Visualizing the performance of SAWEI and baselines (mostly anytime rank or anytime log-regret)


**Usability And Ease Of Reproducibility:**

In general, I believe all experiments are fully reproducible, and all code needed to reproduce figures is available.
However, I also believe that some more effort could have been put into the code submission.
In the current state, installation and running of examples is quite buggy and not fully reproducible.

For example:
* In `setup.py` there is a typo in line 56 (repeated `https`), preventing automatic installation.
* Installation gives a warning `WARNING: dacbo 0.0.1 does not provide the extra 'dev'` due to the code on anonymous GitHub not being a repo (I guess this will be automatically fixed upon release as a proper GitHub repo).
* running `make install-dev` currently fails due to the directory not being a GitHub repo (similarly as above, I believe this will "fix" itself) and therefore the `pre-commit` steps will fail unless one manually performs `git init .` prior to calling `make install-dev`.

Also, I currently cannot run the examples, i.e.

`python awei/optimize.py '+policy/baseline/weischedules=glob(*)' 'seed=range(1,11)' 'instance_set.fid=range(1,25)' 'instance_set.instance=1' 'n_eval_episodes=1' -m` as stated in `AWEI/README.md` will return:

```
Could not override 'instance_set.instance'.
To append to your config use +instance_set.instance=1
Key 'instance_set' is not in struct
    full_key: instance_set
    object_type=dict
```

Similarly, running

`python examples/examples/.py` will fail with:

```
Error executing job with overrides: []
Traceback (most recent call last):
  File "XXX/examples/examples.py", line 11, in main
    printr(OmegaConf.to_container(cfg=cfg, resolve=True))
omegaconf.errors.UnsupportedInterpolationType: Unsupported interpolation type calculate_n_trials
    full_key: smac_kwargs.scenario.n_trials
    object_type=dict
```

I believe there might be a conflict/issue with the hydra setup, but I did not debug any further.

Also, I am not a fan of requiring a slurm setup for many examples - surely, some improvements can be made here.

Finally, the repo structure itself could maybe be improved slightly (e.g., not performing a "manual" installation of SMAC).

Also, benchmark results are not provided so reproducing the figures and tables (without re-running all experiments) is currently not possible (again, I do believe that this will be fixed as stated in the checklist).

---

> ### Author Response · Authors · 2023-05-01
> **Initial Response**
>
> Thank you for your reproducibility review!
> We updated our [repository](https://anon-github.automl.cc/r/SAWEI-9CD6/README.md) according to your review and post the changelog:
>
> - Fix installation
> - Remove example folder and added an example section in README.md (no slurm setup necessary)
> - Add short description of how the interface to BO/SMAC works in README.md
> - Provide link to the data and update plotting scripts
>
> We apologize for the installation issues and updated our installation instructions for ease of use.
>
> In addition we uploaded the data to [here](https://drive.google.com/drive/folders/12jmpJ1VRS3rzRcCd6rrrcjusP19RAmmV?usp=sharing) to produce the plots and updated the notebook for easier reproducing the figures.

---

> > ### Comment · Reproducibility_Reviewer_CqGE · 2023-05-02
> > **Updated Reproducibility Review**
> >
> > Thanks for updating the repository and also providing the raw data!
> > Examples now run smoothly for me and results and figures can in general now be reproduced.
> > I therefore increased my rating.

---

### Official Review · Reviewer_D7zQ · 2023-04-07

**Potential Impact On The Field Of Automl Rating:** 3
**Technical Quality And Correctness Rating:** 3
**Clarity Rating:** 2
**Ethics And Accessibility Rating:** Yes, regarding other reasons (please …

**Summary Of Contributions:**

The paper proposed a self-adjusting acquisition function for BO, based on EI and PI. It compares to a collection of other ways to combine EI and PI on both synthetic and HPO problems.

**Actions Required To Increase Overall Recommendation:**

These are listed in the above sections. In particular, clearing up the difference between the different types of WEI and addressing the concerns in clarity and accessibility.

**Clarity:**

The clarity is generally high, and the paper well-written. The paragraph SAWEI in a Nutshell in section 3 is a good inclusion. Having said this, there are some serious concerns, and later some minor ones.

Serious concerns
- Related work: You say you report findings from enhancing EI to meta-learning neural AFs, but after this sentence claiming you do so, the citations are not referenced to again. You also do not compare to these methods as baselines.
- In eq 1 you define WEI, and say that in your method you adjust its parameter alpha. However, buried in a later paragraph (lines 140-142), you say that you in fact use a different expression for WEI. The one you use should be at least as clear as the one you do not use, and the current presentation is misleading. Also, how does this change make the signal clearer?
- (SAWEI in a nutshell and Algorithm 1): It is unclear how you are tracking the search attitude. Is this done across the optimisation search space or only for the next x selected? You should give these expressions. Does this result in the exploration and exploitation term being made more similar in value? If so, for one point or across points?
- L210-211: you say you can clearly observe the effect of the switching. Where?
- The ablation section is hard to follow. What is an incumbent change and what is a switch? Do you compare all combinations of hyperparameters, or keep the other two constant? It seems like you don’t use the study to choose the values, but to check if they are sensitive, which is reasonable, but not what you say (l232-233). You only do the ablation on one of the data sets and do not use the best-performing combination.

Medium concern
- L75: ‘search attitude’. This is an important concept, so worth defining properly. I take it to be the AFs weighting towards either exploration or exploitation.
- Figure 1 caption: you say the gradient of UBR ‘becomes zero’, but elsewhere you say within a threshold.
- L11 in Algorithm 1 and l166: Is this where the hyperparameter epsilon is used? It’s unclear.
- L170: do you mean that you calculate the mean across seeds within the algorithm or outside it, for evaluation?
- Figure 5 caption: make it clear that this is final performance.
- L276-279. You don’t show this in the paper. Worth referring to the appendix so people know where to find it.
- Limitations: Another limitation is that you show large variation between domains, but only have results for two. I’m not suggesting collecting more results, but acknowledging the limitation. And that you do not compare to more of the related work you mention.
- Limitations (l251-253): your argument seems to be that a simple self-adjusting approach can beat other PI and EI based AFs. The trade-off is that your method is computationally more expensive and more complicated to implement. This suggests that there are times when e.g. EI are better, and times when SAWEI is better. It would strengthen the paper to engage with this.
- In the ablation study figure 6 has different axis scales, making it hard to compare. It doesn’t only matter whether there’s a significant difference, but whether it will lead to a noticeable downstream difference.

The below lists a collection of minor issues which should be easy to correct:
- L14: BBOB should be explained. Write out the acronym.
- L38: I think you mean ‘target value’ instead of ‘cost’. Cost is not used in this sense elsewhere in this paper.
- L44: ‘bells and whistles’: what do you refer to here? Bells and whistles doesn’t let me know, listing examples would be better.
- L84: PI has a weird marker between the P and I.
- L86-88: You swap between referencing Hoffman et al 2014 as GP-Hedge and Portfolio Allocation, reducing clarity.
- L203: General*ly* good performing?
- L207 and Figure 3 caption. Exemplary means very good, you mean example.
- 4.1 BBOB: You don’t refer to 3 b,c,d, only 3a.
- L271: criterion *for* when.
- 264-265: can you be clearer here? The other two components you have mentioned are the initial design and the surrogate model.


**Ethics Details (Optional):**

Some of the figures (2, 3, 4, and 6) have accessibility concerns. Luckily, these should be easy to fix in rebuttal.

- Figure 3 (a,c) has too small a font.
- Figure 2, 3 (b,d) and 4 varies the markers, making it harder to compare the figures. More importantly they are hard to read as the colours are very similar and the linestyles are not varied. Figures 2 and 4 should be replotted so the legend does not overlap the axis label.
- Figure 6 uses a very faint line, making it hard to read. It also uses different axis scales, making it hard to compare the different hyperparameters.


**Overall Review:**

The strengths of the paper are that it proposes a self-adjusting acquisition function, which they compare to a collection of related benchmarks. It is mostly clearly written, tested on a benchmark relevant to the AUTOML community, and engaged with the interesting question of what acquisition function to use.

That being said, there are some important elements that are unclear, and which require fixing. It uses two different versions of WEI, which seems to make the baselines less comparable, and lacks details in how the attitude is tracked. The ablation study and the related work start out with claims of what they do that don’t match up with the rest of the paragraph.

Overall, the paper seems to be almost ready: the plots need some work to be more accessible and the writing needs to be clarified in parts. I expect the authors to be able to remedy most, if not all of these concerns in the rebuttal. And in that case the review will of course increase.


**Potential Impact On The Field Of Automl:**

It could lead to BO having better performance, which again could give better ML models. But this is likely dependent on being integrated in relevant libraries.

It helps to explore the complicated question of acquisition functions, and could also be cited for this.

**Review Confidence:**

3: You are fairly confident in your assessment. It is possible that you did not understand some parts of the submission or that you are unfamiliar with some pieces of related work.

**Review Rating:**

7: Weak Accept: Technically sound paper with moderate-to-high impact and strong evaluation, with perhaps some minor flaws.

**Review Summary:**

It is an interesting paper, relevant to the AUTOML community, but which in its current form is unclear, even misleading in parts, and some details are missing. These parts can be remedied by clarifying the writing and plotting, and I expect to increase the score once this has been done.

**Technical Quality And Correctness:**

There is confusion between two different versions of WEI. It seems that one is used for the baselines and another for the proposed method, which raises the question of whether the improvement comes from this change and not the self-adjusting. Additionally, in table 1 PI* is listed, while in the figure legends it’s given as PI. Which is correct? Why do some of the baselines use PI and others PI*? How can you be sure that this isn’t influencing the results?

---

> ### Author Response · Authors · 2023-04-28
> **Initial Response**
>
> We would like to thank you for your in-depth review, we appreciate it! We will address your comments in the next paragraphs.
> We addressed your low and medium concerns as much as possible in the updated paper version.
> We would like to address your major concerns in our response in more detail.
>
> ## Clarity and Accessibility
>
> We updated the figures for better accessibility and refined the terms of the baselines for clarity. In addition we clear misunderstandings about how we use WEI.
>
> ## PI vs. PI*
>
> Thank you for pointing out the possible confusions between the PI*=WEI($\alpha=1$) and the true PI. The baselines switching from EI to PI after a certain time are not based off WEI. In contrast, the linear step function from $\alpha=0.5$ to $\alpha=1$ and vice versa parametrize WEI, therefore used the notation PI*. Table 1 and the figures both are correct. We updated our notation in the text and the figures.
>
> ## When SAWEI is better
>
> You are right with your suggestions: At times e.g. EI is better and sometimes it is not. For the BBOB functions where we know the function landscape we could link the landscape to the performance of the individual methods. On HPOBench, this is not possible.
> The important thing is that on average SAWEI performs better or similarly with the best baseline. What baseline is best really depends on the problem and this robustness is the strength of our method, i.e. we do not need prior knowledge to obtain solid performance.
>
> ## Alpha Schedules are Different for Different Benchmarks
>
> > L276-279. You don’t show this in the paper. Worth referring to the appendix so people know where to find it.
>
> Thank you for pointing this out. You are right and we added the direct comparison of the $\alpha$ schedules on BBOB and HPOBench to directly compare the different behavior of SAWEI in Figure 5b.
>
> ## Related Work
>
> Thank you for pointing this out. You are right that we missed elaborations on Qin et al (enhancing EI) and Balandat et al. (BoTorch) and Volpp et al. (neural AFs) and we adjusted the text accordingly to include a short summary for both.
>
> ## Different WEI Schedules and Tracking the Search Attitude
>
> We understand that our former presentation of the algorithm can be misleading. We use standard WEI to propose the next configuration to evaluate. However, to determine the search attitude we modify one summand. This modification followed our intuition that the second summand is related to  PI, and PI is more exploitative than (W)EI. Empirically, both methods perform on par (we added a figure in the appendix).
> We updated the text accordingly.
> Regarding tracking the search attitude, we record the summands of WEI for the configuration proposed by WEI to be evaluated next.
> We updated the section "How to Adjust"  and "SAWEI in a Nutshell" to be more explicit on how the search attitude is calculated. Thank you for providing this feedback to improve our paper.
>
> ## Effect of Switching in UBR
>
> The effect of switching from EI to PI is expressed two folds in the UBR: We can see sharp downward bends in the log regret and sharp (often upward) bends in the UBR. This is nicely visible in Appendix Figure 20 (BBOB F13) or Figure 25 (BBOB F18). Same holds for HPOBench, but here log regret often stagnates after the switch. The magnitude of this effect depends on the problem at hand.
> We updated the text to point to the Figure 3(d) and also added a small section in the Appendix showing this on more functions.
>
> ## Ablation Study
>
> We reworded the ablation section to make it easier to follow, and especially reworded L232-233 to reflect our motivation for the sensitivity analysis.
> In particular, we can track the attitude up to the point where the incumbent changed the last time, i.e. until the point where we found a new best performing configuration. Finding a new incumbent is a desirable event in BO and thus lead to this setting.

---

> > ### Comment · Reviewer_D7zQ · 2023-05-02
> > **Thank you for your response**
> >
> > Thank you for your response and the updates to the paper. Especially for the presentation of the search attitude which is now much clearer, and the related work which is much more informative. Figure 6 is also much better now. I still think some of the figure captions are too small, hopefully you can increase them in a camera-ready version. I still think the more clearly presented expression for alpha exploit (eq 4) should be the one you use.
> >
> > Typo: l148-149: You say alpha explore twice, one of them should be exploit.
> >
> > I have improved my score accordingly.

---

### Official Review · Reviewer_2V2m · 2023-04-11

**Potential Impact On The Field Of Automl Rating:** 3
**Technical Quality And Correctness Rating:** 2
**Clarity Rating:** 3

**Summary Of Contributions:**

The paper proposes a way to set a trade-off parameter $\alpha$ in a Bayesian optimization (BayesOpt) policy proposed in previous work.
Specifically, this parameter $\alpha$ controls the behavior of the resulting policy on the exploration/exploitation spectrum of Expected Improvement (EI) -- Probability of Improvement (PI), and the paper proposes to adjust the value of $\alpha$ based on optimization history in the following way.
First, we keep track of a metric called the Upper Bound Regret (UBR), which is the difference between a pessimistic estimate of the incumbent value and a lower bound on the lowest function value.
If this metric grows smaller as optimization progresses, it is an indication that optimization with the current setting is converging, and we might benefit by adjusting the value of $\alpha$.

To update the value of this parameter, we inspect the trade-off between exploration and exploitation in the most recent queries.
If recent queries have favored exploration, we adjust the parameter to bias exploitation and vice versa.
This strategy aims to counteract getting stuck at a local optimum from how we are currently balancing exploration and exploitation.

The authors then present comprehensive experiment results comparing the proposed algorithm against a wide range of benchmarks.

**Actions Required To Increase Overall Recommendation:**

I'd like to understand the exploration term in the acquisition function of SAWEI better, so some discussion on this will be appreciated.
Crucially, more details about how to compare recent exploration vs. exploitation and how the UBR is computed and tuned (by setting $\beta_t$) is needed.

Establishing some theoretical guarantee would help justifying the algorithmic design.
Including stronger baselines such as entropy search policies will make the experiment section more well-rounded.

**Clarity:**

The paper is clear and well-written.
I had an easy time following its arguments and discussions.
In the plots, I sometimes mistake SAWEI, which usually performs well and, Portfolio Allocation, which performs not as well, due to their similar colors.
The authors can consider using different line styles, in addition to colors and markers, to differentiate the benchmarks.

**Overall Review:**

The paper is well-structured and has a nice flow.
The proposed algorithm is natural and easy to understand: it involves adjusting the trade-off parameter in a direction opposite to recent behavior to avoid getting stuck at local optima.

The paper extends a previously proposed BayesOpt policy by designing a way to automatically set its trade-off parameter.
As such, I worry that the contributions are somewhat incremental.

As noted in the **Technical Quality And Correctness** portion, some of the details about the algorithm are not clear or missing from the manuscript.
This makes appreciating the proposed algorithm more difficult.
It's also somewhat disappointing that the paper doesn't mention theoretical guarantees for the proposed policy.
Given its connection to PI and EI, both of which have their own guarantees, I was hoping some results could be derived for SAWEI.

Finally, I believe the experiment section should include other commonly used BayesOpt policies to ensure that there's value in adopting the current approach.

**Potential Impact On The Field Of Automl:**

The paper extends an interesting research direction and can potentially lead to impacts.

**Review Confidence:**

4: You are confident in your assessment, but not absolutely certain. It is unlikely, but not impossible, that you did not understand some parts of the submission or that you are unfamiliar with some pieces of related work.

**Review Rating:**

4: Weak Reject: For instance, a paper with minor technical flaws, limited impact, and/or weak evaluation.

**Review Summary:**

I recommend weak rejection.
While I think this paper proposes an interesting approach to the problem of controlling the behavior of WEI, I fear the contributions are limited and some of the crucial details are missing to make this work a strong submission.

**Technical Quality And Correctness:**

The paper proposes a reasonable approach to the problem at hand.
I would have liked to see more details being included when the algorithm is being developed.
For example, what is the value of $\beta_t$ during the calculation of the UBR metric?
I can imagine the value of $\beta_t$ can greatly affect the behavior of the resulting policy, and the experiment section could include a study on how this value affects performance.

Another point that wasn't clear to me is how the balance between exploration and exploitation is assessed.
On lines 140–144, the authors mention that $\Phi \left( \boldsymbol{z}(\boldsymbol{x}) \right)$ could be used instead of a slightly more complicated term to indicate exploitation and be compared against the exploration term $\hat{s}(\boldsymbol{x}) \phi \left( \boldsymbol{z}(\boldsymbol{x}) \right)$.
It's not obvious to me the two terms have the same units.
As this comparison is at the heart of the algorithm, I would have liked to see more discussion being devoted to this.

The paper suggests that $\alpha = 0$ corresponds to pure exploration, and I'm not sure if this is true.
It's true that keeping everything else constant, $\Phi \left( \boldsymbol{z}(\boldsymbol{x}) \right)$ increases as uncertainty $\hat{s}(\boldsymbol{x})$ increases, but optimizing $\Phi \left( \boldsymbol{z}(\boldsymbol{x}) \right)$ doesn't exactly correspond to optimizing $\hat{s}(\boldsymbol{x})$.
Specifically, the term $\boldsymbol{z}(\boldsymbol{x})$ inside the CDF also assigns higher scores to points with predictive means $\hat{y}(\boldsymbol{x})$ being close to the incumbent value $f_{\text{min}}$.
So maximizing $\Phi \left( \boldsymbol{z}(\boldsymbol{x}) \right)$ might still bias us towards exploitative behavior.
It would be great if there could be further discussions on this and how to potentially encourage true exploration.

I would have liked for the experiment section to include other commonly used policies such as UCB/LCB and entropy search policies, the latter of which have been shown to balance exploration and exploitation in their own information-theoretic way.

The "rank" metric being used in Figures 2 and 4 was also not entirely clear to me.
Are the authors keeping track of the order of the policies in terms of performance across the different problems?
It's not a standard metric in BayesOpt as far as I know, and perhaps the log regret vs. number of iterations plots are sufficiently clear on their own.

---

> ### Author Response · Authors · 2023-04-28
> **Initial Response**
>
> Thank you for reviewing our paper!
>
> ## Figures
>
> We updated our figures for better accessibility.
>
> ## UBR and $\beta_t$
>
> We set $\beta_t = 2 \log (d  t^2 / \beta), \beta = 1$ for UCB/LCB as done in SMAC3 [Lindauer et al, 2022] following the original UCB [Srinivas et al., 2010].
> We added this information in the experimental setup.
> We agree that an analysis on the behavior of UBR depending on $\beta_t$ is interesting work to refine the stopping criterion [Makarova et al., 2022] and our work.
>
> ## Search Attitude
>
> Empirically, we see that both versions, PI and the modified PI (the exploitation-term of WEI), perform almost on par, with PI a little better. We added this as a comment in “How to Adjust” and a small appendix section. The scales of both terms are different due to the weighting of $(\mu(x) - f_{min})$.
> We would also like to stress that we do not denote $\alpha=0$ as pure exploitation, merely as the exploring term in WEI.
>
> ## True Exploration
>
> We agree that WEI does not perform true exploration (even with $\alpha = 0$). Nevertheless, as argued in the original paper it provided a stronger exploration attitude. In principle our method could be extended to linear combinations of other acquisition functions or even to adjust the $\beta$ in UCB to trade-off exploration-exploitation. In the most extreme case, we could also use our approach to adjust the probability of sampling a point based on the acquisition function or from random sampling – we note that for example, SMAC3 uses a fixed probability for that.
>
> ## Inclusion of More Baselines
>
> We included LCB as an additional baseline which does not change our general conclusions.
>
> ## Rank as a Metric
>
> We choose ranks as a metric because comparing the log regret across functions and instances is difficult: Often we see a wide spread in performance and aggregating obscures what we are originally interested in: What method works best on this specific instance?
> For completeness, we have boxplots of the log regrets for each function / task in the appendix.
>
> [Lindauer et al., 2022] Marius Lindauer, Katharina Eggensperger, Matthias Feurer, André Biedenkapp, Difan Deng, Carolin Benjamins, Tim Ruhkopf, René Sass, Frank Hutter:
> SMAC3: A Versatile Bayesian Optimization Package for Hyperparameter Optimization. J. Mach. Learn. Res. 23: 54:1-54:9 (2022)
>
> [Makarova et al., 2022] Makarova, A., Shen, H., Perrone, V., Klein, A., Faddoul, J., Krause, A., Seeger, M., and Archambeau, C. (2022). Automatic termination for hyperparameter optimization. In Guyon, I., Lindauer, M., van der Schaar, M., Hutter, F., and Garnett, R., editors, International Conference on Automated Machine Learning, AutoML 2022, 25-27 July 2022, Johns Hopkins University, Baltimore, MD, USA, volume 188 of Proceedings of Machine Learning Research, pages 7/1–21. PMLR.
>
> [Srinivas et al., 2010] Srinivas, N., Krause, A., Kakade, S., and Seeger, M. (2010). Gaussian process optimization in the bandit setting: No regret and experimental design. In Proc. of ICML’10, pages 1015–1022.

---

> > ### Comment · Reviewer_2V2m · 2023-05-03
> > **remaining questions**
> >
> > I thank the authors for their response.
> > - Regarding the two "versions" of the PI term, my original question was along the lines of, how are we able to balance between the exploration term and the exploitation term in WEI if the two terms aren't in the same units? I agree that the two versions of PI should perform on par since it seems to me they have the same maximizer, but my concern is: (1) is it possible for one term to always be greater than the other, making it impossible to control for the tradeoff?, and (2) how can the user know that the balance is being facilitated effectively?
> > - Thanks for the clarification on pure exploration; that clears up my confusion. Perhaps the text could be updated to indicate that pure exploration isn't being pursued with $\alpha = 0$.
> > - Thanks for adding in LCB. Do you have any thoughts other policies such as entropy search policies and knowledge gradient? I understand your time was limited, but feel free to share any comments.
> > - Any thoughts on theoretical guarantees?

---

> > > ### Author Response · Authors · 2023-05-09
> > > **Answer to Remaining Questions**
> > >
> > > ## PI Term
> > >
> > > Sorry, we misunderstood what you meant. One clarification before we discuss the scale of the terms: We use the normal WEI (Eq. 1) for proposing the next configuration to evaluate.
> > > We ran our method SAWEI with the attitude determined on the original WEI terms, dubbed SAWEI (modPI), and observed that for BBOB it performs on par and on HPOBench the performance is not convincing (see [here](https://imgur.com/a/RZyQJwE)). This is because `a_exploit` (the modulated PI term) is always smaller than a_explore (the EI term) steering $\alpha$ towards exploitation on HPOBench. Using the pure PI as `a_exploit` however rightfully determines the attitudes and steers $\alpha$ towards exploration.
> > > To address your questions, (1) in our results $\alpha$ is adapted in different directions in our experiments, which can be seen in the Appendix plots. (2) Whether the balance is set effectively is difficult to determine online. Until the convergence of BO (with WEI and fixed $\alpha$) SAWEI cannot be worse. Afterwards we can only improve further by now using a different $\alpha$.
> > >
> > >
> > > ## Pure Exploration Wording
> > >
> > > We made it explicit in the text that $\alpha=0$ does not equal pure exploration.
> > >
> > > ## Other Acquisition Functions
> > >
> > > In the experiments we saw that static acquisition functions (e.g., EI, LCB, PI) are strong on different types of problems. We hypothesize that entropy search and knowledge gradient acquisition functions show a similar behavior, i.e., that they work differently well on different kinds of problems but this needs to be empirically validated.
> > >
> > > ## Theoretical Guarantees
> > > We believe that with careful work theoretical guarantees can be derived. For WEI itself there are no guarantees yet, neither for the static nor the dynamic case.
> > > One could think of using theoretical guarantees for EI and PI and superposing them. Factoring in the dynamicity of WEI is not straightforward, one might start with providing guarantees for $\alpha=0$ and $\alpha=1$.
> > > However, SAWEI also depends on the Upper Bound Regret (UBR), to be more precise, on the gradients of the UBR. Makarova et al. (2022) do provide a bound for the target objective in terms of statistical error and the simple regret. Both quantities are not known and estimated. For the statistical error they use a cross-validation estimator which is not applicable in our case as we do not necessarily optimize ML models on a dataset.
> > > A straightforward but unelegant method yielding theoretical convergence guarantees would be interleaving with random samples once in a while, as in Hutter et al. (2011). In the limit, we are then guaranteed to converge to the optimum. A more direct analysis of SAWEI could possibly be done along the lines of  Doerr, Doerr & Yang (2016), which provides performance guarantees for a parameter control scheme using an $\epsilon$-greedy multi-armed bandit approach.
> > >
> > > [Hutter et al., 2011] Hutter, F., Hoos, H. H., & Leyton-Brown, K. (2011). Sequential model-based optimization for general algorithm configuration. In Learning and Intelligent Optimization: 5th International Conference, LION 5, Rome, Italy, January 17-21, 2011. Selected Papers 5 (pp. 507-523). Springer Berlin Heidelberg.
> > >
> > > [Makarova et al., 2022] Makarova, A., Shen, H., Perrone, V., Klein, A., Faddoul, J. B., Krause, A., ... & Archambeau, C. (2022, September). Automatic termination for hyperparameter optimization. In International Conference on Automated Machine Learning (pp. 7-1). PMLR.
> > >
> > > [Doerr, Doerr, Yang, 2016] Doerr, B., Doerr, C., & Yang, J. (2016). k-Bit mutation with self-adjusting k outperforms standard bit mutation. In Parallel Problem Solving from Nature–PPSN XIV: 14th International Conference, Edinburgh, UK, September 17-21, 2016, Proceedings 14 (pp. 824-834). Springer International Publishing.

---

### Official Review · Reviewer_naa8 · 2023-04-12

**Potential Impact On The Field Of Automl Rating:** 3
**Technical Quality And Correctness:** It seems correct, but more justificat…
**Technical Quality And Correctness Rating:** 3
**Clarity Rating:** 2

**Summary Of Contributions:**

This work suggests a new acquisition function, self-adjusting weighted expected improvement, in Bayesian optimization.  To control an ability to explore and exploit, it adds a new parameter to balance both exploration and exploitation.  In particular, such a parameter is adaptively adjusted.  The authors show some experimental results of diverse acquisition functions.

**Actions Required To Increase Overall Recommendation:**

I am on a positive side now.  I would like to hear how you will revise the paper and justify the proposed method.

**Clarity:**

The aspect ratios of figures should be fixed.  Also, some legends in figures are overlapped with graphs and font sizes in figures are too small.

**Overall Review:**

I think that it proposes a novel acquisition function, which is very interesting.  However, the proposed acquisition function is somewhat heuristic.  I think that more detailed justification is required.

* I think that the authors need to do more exhaustive analysis on the proposed method.

* Baselines are unclear.  Please elaborate them.  In particular, please add a description of EI -> PI* (Linear) and PI* -> EI (linear).

* According to the previous work, the upper bound regret is used to determine when we terminate a Bayesian optimization round.  Why can it be employed to adjust exploration and exploitation?

* Could you explain how you implement Line 10 of Algorithm1?

* I think that Figures 3(a) and 3(c) are redundant.  If a space for contents is tight, it is okay to move them to the appendix.

* It can be a preference, but the y-axes of Figures 2, 4, and 5 should be flipped, i.e., higher ranks are upper.

**Potential Impact On The Field Of Automl:**

It would be impactful in the field of AutoML, because it suggests a new acquisition function in Bayesian optimization.

**Review Confidence:**

4: You are confident in your assessment, but not absolutely certain. It is unlikely, but not impossible, that you did not understand some parts of the submission or that you are unfamiliar with some pieces of related work.

**Review Rating:**

6: Borderline Leaning Accept: Technically sound paper where reasons to accept outweigh reasons to reject. Please use sparingly.

**Review Summary:**

Please see the text boxes above.

---

> ### Author Response · Authors · 2023-04-28
> **Initial Response**
>
> Thank you for reviewing our work. In the following we will address your comments.
>
> ## Clarity and Presentation
>
> We updated our figures for better accessibility (new colors, bigger font sizes, no overlaps).
> We keep the rank orientation as is to fit the notions of "lower is better" for the log regret and UBR. We hope you are fine with this.
> In addition we improved the naming of our baselines to avoid confusion.
>
> ##  UBR as a signal to adjust
>
> The upper bound regret (UBR) can be used to stop Bayesian optimization (BO) [Makarova et al., 2022]. This means the UBR signals whether it is worth continuing optimization. We add our intuition that this holds for the *current optimizer settings*. This is empirically supported by observing the UBR for the switching policies (EI to PI) where we see sharp bends in the UBR after switching.
> In our case, *current setting* implicitly describes the search attitude whether it is exploring or exploiting. Our intiuition is that if we converged with a lot more exploration, more exploitation should help next, and vice versa. Therefore we can use the UBR to signal when we should change our search attitude. Just to clarify: We do not use the UBR itself to adjust the exploration-exploitation trade-off.
> We added a short section with this in the Appendix and refer to it in "When To Adjust".
>
> ## Smooth with IQM (Line 10 in Algorithm 1)
>
> The UBR exhibits a rough signal therefore we smooth it as follows: We apply a moving interquartile mean (IQM) (25%-75%, to remove the spikes) with a window size of 7. This is basically the moving average but with IQM. We updated the description of the algorithm accordingly in "SAWEI in a Nutshell".

---

> > ### Comment · Reviewer_naa8 · 2023-05-01
> > **Thank you for your response**
> >
> > Thank you for your response.
> >
> > I acknowledge that I have read your rebuttal.  I will keep my score.

---

### Official Review · Reviewer_7ELw · 2023-04-13

**Potential Impact On The Field Of Automl:** This work can have impact on the sele…
**Potential Impact On The Field Of Automl Rating:** 2
**Technical Quality And Correctness Rating:** 2
**Clarity Rating:** 3
**Actions Required To Increase Overall Recommendation:** Please see the weakness.

**Summary Of Contributions:**

This work proposes an adaptive strategy to adjust the hyper-parameter in weighted Expected Improvement (WEI). The strategy is simple yet efficient.

**Clarity:**

I have one concern about generalizability. This method aims to adjust the hyper-parameter of the weighted EI acquisition function adaptively, and it can degrade to eighter pure exploration, EI, or PI. I am afraid that this method cannot integrate other acquisition functions, such as UCB, TS, Knowledge gradient, etc.

**Overall Review:**

Strength:

++The proposed SAWEI is simple yet efficient.

Weakness:

-- I have one concern about generalizability. This method aims to adjust the hyper-parameter of the weighted EI acquisition function adaptively, and it can degrade to eighter pure exploration, EI, or PI. I am afraid that this method cannot integrate other acquisition functions, such as UCB, TS, Knowledge gradient, etc.

-- I have concerns about the settings of the hyper-parameter.

1. Why did the authors fix the $\Delta \alpha$? How about adjusting it according to the UBR?

2. According to Fig. 6(a), it seems that $\epsilon=0.05$ has the best performance, then why the authors set  $\epsilon=0.1$ by default (in Line 166)?



**Review Confidence:**

3: You are fairly confident in your assessment. It is possible that you did not understand some parts of the submission or that you are unfamiliar with some pieces of related work.

**Review Rating:**

4: Weak Reject: For instance, a paper with minor technical flaws, limited impact, and/or weak evaluation.

**Review Summary:**

I am afraid that this method should be further refined so as to integrate more rich acquisition functions. Moreover, the settings of $\epsilon$ and $\Delta \alpha$ might be kind of groundless.

**Technical Quality And Correctness:**

The overall method is kind of sound. However, I have concerns about the settings of the hyper-parameter.

1.	Why did the authors fix the $\Delta \alpha$? How about adjusting it according to the UBR?

2.  According to Fig. 6(a), it seems that $\epsilon=0.05$ has the best performance, then why the authors set  $\epsilon=0.1$ by default (in Line 166)?

---

> ### Author Response · Authors · 2023-04-28
> **Initial Response**
>
> Thank you for your review. In the following we address your concerns about SAWEI’s hyperparameters and generalizability.
>
> ## Setting of SAWEI’s Hyperparameters
>
> Regarding the hyperparameter settings:
>
> [ $\Delta\alpha$] We agree that setting $\Delta\alpha$ as a function of the UBR proposes an elegant solution.
> However, with a fixed $\Delta\alpha$ we are already able to determine  well-performing $\alpha$ schedules per problem at hand. Therefore, we leave this as a future work to further improve SAWEI.
>
> [$\epsilon$] This is a good spot. However we can only view those results in the context of the ablation benchmark which was BBOB. The chosen $\epsilon$ performs well on both inherently different benchmarks, BBOB and HPOBench. Since these are our results on new test functions (not used for development), we cannot tune SAWEI’s hyperparameters on that without risking overfitting and thus providing over-optimistic empirical results.
>
> ## Generalizability
>
> There are two core ideas behind this work. First, we want to adjust the exploration-exploitation trade-off *online*. Second, we want to adjust it whenever we converge with the current settings. The idea of adjusting the weight of a linear combinations not exclusive to WEI: We can think of linearly combining *any* two acquisition functions, e.g. UCB or TS. We chose WEI because its summands can be attributed to explorative or exploitative behavior. With the first goal in mind, to adjust the trade-off, other acquisition functions need to be chosen with care.
>
> In addition, SAWEI exhibits two different attitudes on the two suites that we consider (please see the comparison of $\alpha$ schedules in the newly added Figure 5b and paragraph in 4.1 ). This (1) motivates the need for a dynamic and self-adjusting AF, (2) is interpretable given the very different nature of the two summands (more exploitative/more explorative) and (3) renders it likely that SAWEI is applicable on a multitude of domains.

---

### Official Review · Reviewer_z9vf · 2023-04-13

**Potential Impact On The Field Of Automl Rating:** 2
**Technical Quality And Correctness Rating:** 3
**Clarity:** This paper is well-orgnized and well-…
**Clarity Rating:** 4
**Actions Required To Increase Overall Recommendation:** 1) More experiments on some non-synth…

**Summary Of Contributions:**

This paper focuses on the research problem of self-adjusting hyper-parameters of Bayesian Optimization (BO) to balance the exploration and exploitation.
The contibutions of this paper are:
1. Clearly formulation of the problem, i.e., when and how should we adjust the hyper-parameters.

2. The proposed method can self-adjust the important hyper-parameter of acquisition function.

**Overall Review:**

Pros

1. This paper is well-motivated and well-written.

Cons
1. The advantages of the current method is not significant enough. For example, the final rank of the proposed method is over 4 (of 11) on two types of test-beds.

2. More issues please refer to "Actions Required To Increase Overall Recommendation".

**Potential Impact On The Field Of Automl:**

The research problem (DAC of BO) is interesting and important.
However, the proposed method is not novel enough.

**Review Confidence:**

4: You are confident in your assessment, but not absolutely certain. It is unlikely, but not impossible, that you did not understand some parts of the submission or that you are unfamiliar with some pieces of related work.

**Review Rating:**

7: Weak Accept: Technically sound paper with moderate-to-high impact and strong evaluation, with perhaps some minor flaws.

**Review Summary:**

This paper is well-motivated and well-written. The research problem is important. However, the proposed method is not novel and the experimental results are weak.

**Technical Quality And Correctness:**

This paper is kind of heuristic and seems technical right. However, the lacking of theoretical analysis make it not solid enough.

---

> ### Author Response · Authors · 2023-04-28
> **Initial Response**
>
> Thank you for your review. We would like to address your comments in the following paragraphs.
>
> ## Significance
>
> > The advantages of the current method is not significant enough. For example, the final rank of the proposed method is over 4 (of 11) on two types of test-beds.
>
> This is because the performance depends on the specific landscape and it is quite unlikely to have an optimal search attitude for each problem/instance - and this is not even what we aim for. We show that even with an average rank of 4 on BBOB, our method works out of the box and is on par with other strong baselines. So, for practitioners with no knowledge about the black-box landscape (as the name black-box suggests) do not know anything about our problem structure, SAWEI is a very robust and adaptive choice, not only w.r.t. final performance but also anytime performance.
> This is especially apparent in the results on HPOBench (Figure 4)
>
> ## Experiments
>
> > More experiments on some non-synthetic tasks would further improve this work.
>
> We emphasize that we evaluated SAWEI on 5 different models with different configuration spaces on 8 datasets. So, overall we show results on 40 HPO tasks. We believe that this is much more than most papers on BO provide as evidence and is considered a very thorough analysis on non-synthetic benchmarks.
>
> ## More Baselines / Multi-Armed Bandits
>
> We note that we included a multi-armed bandit-based method which is Portfolio Allocation [Hoffman et al., 2011]. There, they have a portfolio of nine acquisition functions (three parametrizations of each EI, PI and LCB). They optimize their portfolio with their algorithm GP-Hedge, adjusting the selection probabilities (which acquisition function is used to determine the next configuration) by the mean of the selected configuration. We note that although we consider only the adaptation of a single acquisition function (WEI), we clearly outperform GP-Hedge.
> To add even more baselines, we also evaluated the well-known LCB that is also outperformed by SAWEI.
>
> ## Comparison with RL-based method
>
> While RL promises more tailored solutions, it is not an easy-to-integrate baseline. First, RL needs to be trained offline and the number of required data points is huge compared to the offline-free, on-the-fly adaptation of SAWEI. From our own experiences, we expect that RL approaches require thousands or even millions of samples to perform well and robustly. In view of that each sample in BO refers to a training of machine learning algorithm, this is often not feasible in practice.  In addition, it is not clear from the beginning what are good state features to describe the state of BO. Although we see promise in applying RL-based methods, comparing against DAC-BO is out of scope of this paper.
>
> The strong point of our method is that a simple idea, working adaptively on the fly with no pretraining necessary, yields strong performance.

---

> > ### Comment · Reviewer_z9vf · 2023-05-02
> > **Response to authors**
> >
> > Thanks for your replies. I will increase my score.

---

### Author Response · Authors · 2023-04-28
**Central Response**

We thank all reviewers for their thoughtful and valuable feedback. We address individual comments in direct replies to the reviews.
As a result of the feedback received we made several updates to the paper:

- Added LCB as another baseline
- Improved markers and colors for Figures 2, 3 (b,d), 4
- Updated Figure 6 (Hyperparameter Sensitivity) (same scale, better font sizes)
- Reworded ablation study
- Refined description of search attitude
- Moved Appendix section “How to Detect Adjustment Point” to “SAWEI in a Nutshell”, better describes usage of hyperparameter $\epsilon$
- Improved names for baselines, i.e. the modulated PI (PI*) is now denoted as WEI($\alpha=1$) to avoid confusion
- Add short appendix section to show that UBR changes after changing optimizer settings (e.g. switching acquisition function)
- Add Figure 5b and description to directly compare the different behavior of SAWEI on BBOB and HPOBench in terms of $\alpha$
- Rephrased beginning of “Related Work”

We marked the updated text with green color.